# Sliced Mutual Information:
# A Scalable Measure of Statistical Dependence

**Ziv Goldfeld**
Cornell University
goldfeld@cornell.edu

**Kristjan Greenewald**
MIT-IBM Watson AI Lab
kristjan.h.greenewald@ibm.com

## Abstract

Mutual information (MI) is a fundamental measure of statistical dependence, with a myriad of applications to information theory, statistics, and machine learning. While it possesses many desirable structural properties, the estimation of high-dimensional MI from samples suffers from the curse of dimensionality. Motivated by statistical scalability to high dimensions, this paper proposes *sliced* MI (SMI) as a surrogate measure of dependence. SMI is defined as an average of MI terms between one-dimensional random projections. We show that it preserves many of the structural properties of classic MI, while gaining scalable computation and efficient estimation from samples. Furthermore, and in contrast to classic MI, SMI can grow as a result of deterministic transformations. This enables leveraging SMI for feature extraction by optimizing it over processing functions of raw data to identify useful representations thereof. Our theory is supported by numerical studies of independence testing and feature extraction, which demonstrate the potential gains SMI offers over classic MI for high-dimensional inference.

## 1 Introduction

Mutual information (MI) is a universal measure of dependence between random variables, defined as

$$\mathsf{I}(X;Y) := \int_{\mathcal{X} \times \mathcal{Y}} \log \left( \frac{\mathrm{d}P_{X,Y}}{\mathrm{d}P_X \otimes P_Y} \right) \mathrm{d}P_{X,Y}, \tag{1}$$

where $(X, Y) \sim P_{X,Y}$ and $\mathrm{d}P/\mathrm{d}Q$ is the Radon-Nikodym derivative of $P$ with respect to (w.r.t.) $Q$. It possesses many desirable properties, such as meaningful units (bits or nats), nullification if and only if (iff) $X$ and $Y$ are independent, convenient variational forms, and invariance to bijections. In fact, MI can be obtained axiomatically as a unique (up to a multiplicative constant) functional satisfying several natural 'informativeness' conditions [1]. As such, it found a variety of applications in communications, information theory, and statistics [1, 2]. More recently, it was adopted as a figure of merit for studying [3, 4, 5, 6, 7, 8] and designing [9, 10, 11, 12, 13] machine learning models.

MI is a functional of the joint distribution $P_{X,Y}$ of the considered random variables. In practice, this distribution is often not known and only samples from it are available, thereby necessitating estimation of $\mathsf{I}(X;Y)$. While this topic has received considerable attention [14, 15, 16, 17, 18], MI is fundamentally hard to estimate in high-dimensional settings due to an exponentially large (in dimension) sample complexity [19]. Motivated by statistical efficiency in high dimensions and inspired by recent slicing techniques of statistical divergences [20, 21, 22, 23], this paper introduces *sliced* MI (SMI) as a surrogate notion of informativeness. We show that SMI inherits many of the properties of classic MI, while allowing for efficient estimation. Furthermore, in certain aspects, SMI is more compatible with modern machine learning practice than classic MI. In particular, deterministic transformations of the random variables can increase SMI, e.g., if the resulting codomains have more informative slices (in classic MI sense ). This enables using SMI as a figure of merit for feature extraction by identifying transformation (e.g., NN parameters) that maximize it.

35th Conference on Neural Information Processing Systems (NeurIPS 2021).

## 1.1 Contributions

SMI is defined as the average of MI terms between one-dimensional random projections. Namely, if $\mathbb{S}^{d-1}$ denotes the $d$-dimensional sphere (whose surface area is designated by $S_{d-1}$), we define

$$\mathsf{SI}(X;Y) := \frac{1}{S_{d_x-1}S_{d_y-1}} \oint_{\mathbb{S}^{d_x-1}} \oint_{\mathbb{S}^{d_y-1}} \mathsf{I}(\theta^\mathsf{T}X; \phi^\mathsf{T}Y)\mathrm{d}\theta\mathrm{d}\phi. \tag{2}$$

We may similarly define a max-sliced version by maximizing over projection directions, as opposed to averaging over them. Despite the projection of $X$ and $Y$ to a single dimension, SMI preserves many of the properties of MI. For instance, we show that SMI nullifies iff random variables are independent, it satisfies a chain rule, can be represented as a reduction in sliced entropy, admits a variational form, etc. Further, SMI between (jointly) Gaussian variables is tightly related to their canonical correlation coefficient [24]. This is in direct analogy to the relation between classic MI of Gaussian variables $(X, Y)$ and their correlation coefficient, where $\mathsf{I}(X;Y) = -\frac{1}{2}\log\left(1 - \rho^2(X,Y)\right)$.

SMI is well-positioned for statistical estimation in high-dimensional setups, where one estimates $\mathsf{SI}(X;Y)$ from $n$ i.i.d. samples of $P_{X,Y}$. While the error of standard MI (or entropy) estimates, e.g., those in [16, 25, 26], scales as $n^{-1/d}$ when $d$ is large, the same estimators admit statistical efficiency when $d = 1, 2$, converging at (near) parametric rates. Combining such estimators with Monte-Carlo (MC) sampling to approximate the integral over the unit sphere, we prove that the overall error scales (up to log factors) as $m^{-1/2} + n^{-1/2}$, where $m$ is the number of MC samples and $n$ is the size of the high-dimensional dataset. We validate our theory on synthetic experiments, demonstrating that SMI is a scalable alternative to classic MI when dealing with high-dimensional data.

A notable contrast between classic and sliced MI involves the data processing inequality (DPI). Classic MI cannot grow as a result of processing the involved variables, namely, $\mathsf{I}(X;Y) \geq \mathsf{I}\left(f(X);Y\right)$ for any deterministic function $f$. This is since MI encodes arbitrarily fine details about $(X, Y)$ as variables in the ambient space, and transforming them cannot reveal anything that was not already there. SMI, on the other hand, only considers one-dimensional projections of $X$ and $Y$, some of which can be more correlated than others. Consequently, SMI can grow as a result of deterministic transformations, i.e., $\mathsf{SI}(X;Y) < \mathsf{SI}\left(f(X);Y\right)$ is possible if *projections* of $f(X)$ are more informative about *projections* of $Y$ than those of $X$ itself. We show theoretically and demonstrate empirically that SMI is increased by projecting the data on more informative directions, highlighting its compatibility with feature extraction tasks.

## 2 Preliminaries and Background

We take $\mathcal{P}(\mathbb{R}^d)$ as the class of all Borel probability measures on $\mathbb{R}^d$. Elements of $\mathcal{P}(\mathbb{R}^d)$ are denoted by uppercase letters, with subscripts to indicate the associated random variables, e.g., $P_X$ or $P_{X,Y}$. The support of $P_X$ is $\mathrm{supp}(P_X)$. Our focus throughout is on absolutely continuous random variables; we use lowercase letters, such as $p_X$ or $p_{X,Y}$, to denote probability density functions (PDFs). For a function $f : \mathbb{R}^d \to \mathbb{R}^{d'}$ and a distribution $P_X \in \mathcal{P}(\mathbb{R}^d)$, we write $f_\sharp P_X$ for the pushforward measure of $P_X$ through $f$, i.e., $f_\sharp P_X(A) = P_X\left(f^{-1}(A)\right)$. The $L^p(\mathbb{R}^d)$ norm of $f$ is denoted by $\|f\|_{p,d}$. The $d$-dimensional unit sphere is $\mathbb{S}^{d-1}$, and its surface area is $S_{d-1} = 2\pi^{d/2}/\Gamma(d/2)$, with $\Gamma$ as the gamma function. We also define slicing along $\theta$ as $\pi^\theta(x) := \theta^\mathsf{T}x$.

**Mutual information and entropy.** Information measures, such as MI and entropy, are ubiquitous in information theory and machine learning. MI is defined in (1) and can be equivalently written in terms of the Kullback-Leibler (KL) divergence as $\mathsf{I}(X;Y) := \mathsf{D_{KL}}\left(P_{X,Y} \big\| P_X \otimes P_Y\right)$. $\mathsf{I}(X;Y)$ thus quantifies how far, in the KL sense, $(X,Y) \sim P_{X,Y}$ are from being independent. The differential entropy of a continuous random variable $X$ with density $p_X$ is $\mathsf{H}(X) := -\mathbb{E}\left[\log\left(p_X(X)\right)\right]$, quantifying a notion of uncertainty associated with $X$. For a pair $(X,Y) \sim P_{X,Y}$, the conditional entropy of $X$ given $Y$ is $H(X|Y) := \int_\mathcal{Y} \mathsf{H}(X|Y=y)\mathrm{d}P_Y(y)$, where $\mathsf{H}(X|Y=y)$ is computed w.r.t. $P_{X|Y=y}$. Conditional MI is similarly defined as $\mathsf{I}(X;Y|Z) := \int_\mathcal{Z} \mathsf{I}(X;Y|Z=z)\mathrm{d}P_Z(z)$. With these definitions, one can represent MI as[1] $\mathsf{I}(X;Y) = \mathsf{H}(X) - \mathsf{H}(X|Y) = \mathsf{H}(Y) - \mathsf{H}(Y|X)$, thus interpreting MI as the reduction in the uncertainty regarding one variable as a result of observing the other. Another useful decomposition is the MI chain rule $\mathsf{I}(X,Y;Z) = \mathsf{I}(X;Z) + \mathsf{I}(Y;Z|X)$.

---

[1] Assuming that the appropriate PDFs exist.

**Data processing inequality.** The DPI states that $\mathsf{I}(X;Y) \geq \mathsf{I}(X;Z)$, if $X \leftrightarrow Y \leftrightarrow Z$ forms a Markov chain. This inequality is a cornerstone for many information theoretic derivations and is natural when there are no computational restrictions on the model. However, given a restricted computational budget, processing the input may aid inference. Deep neural network classifiers are an excellent example: they generate a hierarchy of processed representations of the input that are increasingly useful (although not more informative in the Shannon MI sense) for inferring the label. The incompatibility between the DPI and deep learning practice was previously observed in [27], motivating their definition of a computationally restricted MI variant that can be grow from processing. As we show in Section 3.2, SMI also shares this property.

## 3 Sliced Mutual Information

Our goal is to define a surrogate notion of MI that is more scalable for computation and estimation from samples in high dimensions. We propose SMI as defined next.

**Definition 1** (Sliced MI). *Fix* $(X,Y) \sim P_{X,Y} \in \mathcal{P}(\mathbb{R}^{d_x} \times \mathbb{R}^{d_y})$. *Let* $\Theta \sim \mathsf{Unif}(\mathbb{S}^{d_x-1})$ *and* $\Phi \sim \mathsf{Unif}(\mathbb{S}^{d_y-1})$ *be independent of each other and of* $(X,Y)$. *The SMI between* $X$ *and* $Y$ *is*

$$\mathsf{SI}(X;Y) := \mathsf{I}(\Theta^\mathsf{T} X; \Phi^\mathsf{T} Y | \Theta, \Phi) = \frac{1}{S_{d_x-1} S_{d_y-1}} \oint_{\mathbb{S}^{d_x-1}} \oint_{\mathbb{S}^{d_y-1}} \mathsf{I}(\theta^\mathsf{T} X; \phi^\mathsf{T} Y) \mathrm{d}\theta \mathrm{d}\phi. \quad (3)$$

Evidently, the SMI between two high-dimensional variables is defined as an average of MI terms between their one-dimensional projections. By the DPI, $\mathsf{SI}(X;Y) \leq \mathsf{I}(X;Y)$ so we inherently introduce information loss. Nevertheless, we will show that SMI inherits key properties of MI such as discrimination between dependence and independence, chain rule, entropy decomposition, etc.

**Remark 1** (One-dimensional variables). *If* $d_x = 1$ *then* $\mathbb{S}^0 = \{\pm 1\}$ *and we have* $\mathsf{SI}(X;Y) = \mathsf{I}(X; \Phi^\mathsf{T} Y | \Phi)$, *which follows by invariance of MI to bijections and the independence of* $(X,Y,\Phi)$ *and* $\Theta$. *Similarly, when* $d_x = d_y = 1$ *we have* $\mathsf{SI}(X;Y) = \mathsf{I}(X;Y)$.

**Remark 2** (Single projection direction). *When slicing statistical divergences, like the Wasserstein distance [20], one typically considers a single slicing direction. Namely, given that* $X$ *and* $Y$ *are of the same dimension* $d$, *they are both projected onto the same* $\theta \in \mathbb{S}^{d-1}$ *and the distances between* $\theta^\mathsf{T} X$ *and* $\theta^\mathsf{T} Y$ *are then averaged over the unit sphere. While this approach is possible also in the context of SMI, we chose to define it using two slicing directions,* $\theta$ *and* $\phi$, *for several reasons:*

1. *this definition is invariant to rotations of the spaces in which* $X$ *and* $Y$ *take values—with a single direction, rotating either space would change the SMI, seemingly an undesirable property for an information measure;*

2. *it gives rise to a crucial property of SMI, that* $\mathsf{SI}(X;Y) = 0 \iff X$ *and* $Y$ *are independent (see Proposition 1), which does not hold with a single slicing direction;[2]*

3. *it fares naturally with variables of different dimensions (although one can use zero padding to circumvent this issue in the single-direction version); and*

4. *it is inspired by the canonical correlation coefficient [24], that also uses two projection directions.*

To later establish a chain rule and entropy-based decompositions, we define SMI between more than two random variables, conditional SMI, and sliced entropy.

**Definition 2** (Joint and conditional SMI). *Let* $(X,Y,Z) \sim P_{X,Y,Z} \in \mathcal{P}(\mathbb{R}^{d_x} \times \mathbb{R}^{d_y} \times \mathbb{R}^{d_z})$ *and take* $\Theta \sim \mathsf{Unif}(\mathbb{S}^{d_x-1})$, $\Phi \sim \mathsf{Unif}(\mathbb{S}^{d_y-1})$, *and* $\Psi \sim \mathsf{Unif}(\mathbb{S}^{d_z-1})$ *mutually independent. The SMI between* $(X,Y)$ *and* $Z$ *is defined as*

$$\mathsf{SI}(X,Y;Z) := \mathsf{I}(\Theta^\mathsf{T} X, \Phi^\mathsf{T} Y; \Psi^\mathsf{T} Z | \Theta, \Phi, \Psi). \quad (4a)$$

*The conditional SMI between* $X$ *and* $Y$ *given* $Z$ *is*

$$\mathsf{SI}(X;Y|Z) := \mathsf{I}(\Theta^\mathsf{T} X; \Phi^\mathsf{T} Y | \Theta, \Phi, \Psi, \Psi^\mathsf{T} Z). \quad (4b)$$

---

[2]Indeed, let $X_1, X_2 \sim \mathcal{N}(0,1)$ be independent, set $X = (X_1, X_2)^\mathsf{T}$ and $Y = (X_2, -X_1)^\mathsf{T}$. As 2-dimensional vectors, $X$ and $Y$ are dependent, but one readily verifies that $\mathsf{cov}(\theta^\mathsf{T} X, \theta^\mathsf{T} Y) = 0$, for any $\theta \in \mathbb{S}^1$. This implies independence of $\theta^\mathsf{T} X$ and $\theta^\mathsf{T} Y$, hence the single slicing direction SMI would nullify in this case.

The expression in (4a) extends Definition 1. Conditional SMI is the sliced information given access to another projected random variable along with its projection direction. Accordingly, conditional SMI is in the spirit of the original definition of $\mathsf{SI}(X; Y)$, incorporating only projected data without introducing additional uncertainty about the direction.

**Remark 3** (Extensions). *Joint and conditional SMI have natural multivariate extensions. For example, $\mathsf{SI}(X_1, \ldots, X_n; Y_1, \ldots, Y_m) := \mathsf{I}(\Theta_1^\intercal X_1, \ldots, \Theta_n^\intercal X_n; \Phi_1^\intercal Y_1, \ldots, \Phi_m^\intercal Y_m | \Theta, \Phi)$, where $\Theta = (\Theta_1 \ldots \Theta_n)$ and $\Phi = (\Phi_1 \ldots \Phi_m)$. The extension of conditional SMI is similar.*

**Definition 3** (Sliced entropy). *Let $(X, Y) \sim P_{X,Y} \in \mathcal{P}(\mathbb{R}^{d_x} \times \mathbb{R}^{d_y})$ and take $\Theta \sim \mathsf{Unif}(\mathbb{S}^{d_x-1})$ and $\Phi \sim \mathsf{Unif}(\mathbb{S}^{d_y-1})$ to be independent. The sliced entropy of $X$ is $\mathsf{SH}(X) := \mathsf{H}(\Theta^\intercal X | \Theta)$, while the conditional sliced entropy of $X$ given $Y$ is $\mathsf{SH}(X|Y) := \mathsf{H}(\Theta^\intercal X | \Theta, \Phi, \Phi^\intercal Y)$.*

Sliced entropy is interpreted as the average uncertainty in one-dimensional projections of the considered random variable. Conditional sliced entropy is the remaining uncertainty when a projected version and the projection direction of another random variable is revealed.

The following proposition shows that SMI retains many of the properties of classic MI.

**Proposition 1** (SMI properties). *The following properties hold:*

1. ***Non-negativity:*** $\mathsf{SI}(X; Y) \geq 0$ *with equality iff $X$ and $Y$ are independent.*

2. ***Bounds:*** $\inf_{\substack{\theta \in \mathbb{S}^{d_x-1} \\ \phi \in \mathbb{S}^{d_y-1}}} \mathsf{I}(\theta^\intercal X; \phi^\intercal Y) \leq \mathsf{SI}(X; Y) \leq \sup_{\substack{\theta \in \mathbb{S}^{d_x-1} \\ \phi \in \mathbb{S}^{d_y-1}}} \mathsf{I}(\theta^\intercal X; \phi^\intercal Y) \leq \mathsf{I}(X; Y).$

3. ***KL divergence:*** *We have $\mathsf{SI}(X; Y) = \mathbb{E}_{\Theta, \Phi}\left[ \mathsf{D}_{\mathsf{KL}}\big( (\pi^\Theta, \pi^\Phi)_\sharp P_{X,Y} \big\| \pi^\Theta_\sharp P_X \otimes \pi^\Phi_\sharp P_Y \big) \right].$*

4. ***Chain rule:*** *For any random variables $X_1, \ldots, X_n, Y, Z$, we have the decomposition $\mathsf{SI}(X_1, \ldots, X_n; Y) = \mathsf{SI}(X_1; Y) + \sum_{i=2}^n \mathsf{SI}(X_i; Y | X_1, \ldots, X_{i-1})$. In particular, $\mathsf{SI}(X, Y; Z) = \mathsf{SI}(X; Z) + \mathsf{SI}(Y; Z | X).$*

5. ***Tensorization:*** *Suppose that $(X_1, Y_1), \ldots, (X_n, Y_n)$ are mutually independent. Then $\mathsf{SI}(X_1, \ldots, X_n; Y_1, \ldots, Y_n) = \sum_{i=1}^n \mathsf{SI}(X_i; Y_i).$*

The proof of Proposition 1 is given in Supplement A.1.

**Remark 4** (SMI versus MI). *Proposition 1 shows that SMI inherits many of the favorable properties of classic MI. Nevertheless, we stress that SMI is posed as a new measure of dependence that (although closely related) is different from MI. In particular, the gap between MI and SMI may not be bounded.[3] SMI thus should not be treated as a proxy of MI, but rather as an alternative figure of merit. The premise of the SMI framework is that its meaningful structure does not translate into computational or statistical inefficiency. Indeed, Section 3.1 shows that SMI can be efficiently estimated with parametric rate (up to logarithmic factors).*

Similarly to MI, the sliced version simplifies when the variables are jointly Gaussian.

**Example 1** (Gaussian SMI). *If $X \sim \mathcal{N}(0, \Sigma_X)$ and $Y \sim \mathcal{N}(0, \Sigma_Y)$ are jointly Gaussian with cross-covariance $\Sigma_{XY}$, then*

$$\mathsf{SI}(X; Y) = \frac{1}{2 S^{d_x-1} S^{d_y-1}} \oint_{\mathbb{S}^{d_x-1}} \oint_{\mathbb{S}^{d_y-1}} \log\left( \frac{1}{1 - \rho^2(\theta^\intercal X, \phi^\intercal Y)} \right) \mathrm{d}\theta \mathrm{d}\phi,$$

*where $\rho(\theta^\intercal X, \phi^\intercal Y) := \frac{\theta^\intercal \Sigma_{X,Y} \phi}{(\theta^\intercal \Sigma_X \theta \phi^\intercal \Sigma_Y \phi)^{1/2}}$ is the correlation coefficient of $\theta^\intercal X$ and $\phi^\intercal Y$. Denoting by $\rho_{\mathsf{CCA}}(X, Y) := \sup_{(\theta, \phi) \in \mathbb{S}^{d_x-1} \times \mathbb{S}^{d_y-1}} \rho(\theta^\intercal X, \phi^\intercal Y)$ the canonical correlation coefficient, we get $\mathsf{SI}(X; Y) \leq -0.5 \log\big(1 - \rho^2_{\mathsf{CCA}}(X, Y)\big).$*

The Gaussian distribution is also special for sliced entropy, where, as for classic entropy, it maximizes $\mathsf{SH}(X)$ under a fixed (mean and) covariance constraint.

**Proposition 2** (Gaussian maximizes sliced entropy). *Let $\mathcal{P}_1(\mu, \Sigma) := \big\{ P \in \mathcal{P}(\mathbb{R}^d) : \mathrm{supp}(P) = \mathbb{R}^d, \mathbb{E}_P[X] = \mu, \mathbb{E}\big[(X - \mu)(X - \mu)^\intercal\big] = \Sigma \big\}$. Then $\arg\max_{P \in \mathcal{P}_1(\mu, \Sigma)} \mathsf{SH}(P) = \mathcal{N}(\mu, \Sigma)$, i.e., the normal distribution maximizes sliced entropy inside $\mathcal{P}_1(\mu, \Sigma)$.*

---

[3]See the Example from the beginning of Section 3.2 and note that under that setup, for any $0 < a < \infty$, we have $\mathsf{I}\big(g_a(X); Y\big) = \infty$ while $\mathsf{SI}\big(g_a(X); Y\big)$ is finite.

The proposition is proven in Supplement A.2, where two additional max-entropy claims are established. Specifically, we show that (i) the uniform distribution on the sphere maximizes sliced entropy over measures supported inside a ball; and (ii) the symmetric multivariate Laplace distribution [28] is the maximizer subject to mean constraints on the $d$-dimensional variable and its projections.

Lastly, SMI admits a variational form in the spirit of the Donsker-Varadhan representation of MI.

**Proposition 3** (Variational form). *Let $\Theta \sim \mathsf{Unif}(\mathbb{S}^{d_x-1})$, $\Phi \sim \mathsf{Unif}(\mathbb{S}^{d_y-1})$ be independent of each other and of $(X, Y) \sim P_{X,Y} \in \mathcal{P}(\mathbb{R}^{d_x} \times \mathbb{R}^{d_y})$, and set $(\tilde{X}, \tilde{Y}) \sim P_X \otimes P_Y$. We have*

$$\mathsf{SI}(X;Y) = \sup_{g: \mathbb{S}^{d_x-1} \times \mathbb{S}^{d_y-1} \times \mathbb{R}^2 \to \mathbb{R}} \mathbb{E}\big[g(\Theta, \Phi, \Theta^\intercal X, \Phi^\intercal Y)\big] - \log\left(\mathbb{E}\left[e^{g(\Theta, \Phi, \Theta^\intercal \tilde{X}, \Phi^\intercal \tilde{Y})}\right]\right),$$

*where the supremum is over all measurable functions for which both expectations are finite.*

This representation is leveraged in Section 4.3 to implement a feature extractor based on SMI neural estimation. The proof is found in Section A.3 of the supplement.

## 3.1 Estimation

A main virtue of SMI is that its estimation from samples is much easier than classic MI. One may combine any MI estimator between scalar variables with an MC integrator to estimate SMI between high-dimensional variables without suffering from the curse of dimensionality. This gain is expected as SMI is defined as an average of low dimensional MI terms.

For $P_{A,B} \in \mathcal{P}(\mathbb{R} \times \mathbb{R})$, let $(A_1, B_1), \ldots, (A_n, B_n)$ be pairwise i.i.d. from $P_{A,B}$. Consider an MI estimator $\hat{\mathsf{I}} : \mathcal{A}^n \times \mathcal{B}^n \to \mathbb{R}_{\geq 0}$ that attains $\delta(n)$ absolute error uniformly over a class $\mathcal{F}$ of distributions:

$$\sup_{P_{A,B} \in \mathcal{F}} \mathbb{E}\Big[\big|\hat{\mathsf{I}}(A^n, B^n) - \mathsf{I}(A;B)\big|\Big] \leq \delta(n). \tag{5}$$

We use $\hat{\mathsf{I}}$ to construct an estimator of SMI. Given high-dimensional pairwise i.i.d. samples $(X_1, Y_1), \ldots, (X_n, Y_n)$ from $P_{X,Y} \in \mathcal{P}(\mathbb{R}^{d_x} \times \mathbb{R}^{d_y})$, first note that $(\theta^\intercal X_i, \phi^\intercal Y_i)$, for $\theta \in \mathbb{S}^{d_x-1}$ and $\phi \in \mathbb{S}^{d_y-1}$, is distributed according to $(\pi^\theta, \pi^\phi)_\sharp P_{X,Y}$. Thus, we can convert $\{(X_i, Y_i)\}_{i=1}^n$ into pairwise i.i.d. samples of the projected variables. Let $\Theta_1, \ldots, \Theta_m$ and $\Phi_1, \ldots, \Phi_m$ be i.i.d. according to $\mathsf{Unif}(\mathbb{S}^{d_x-1})$ and $\mathsf{Unif}(\mathbb{S}^{d_y-1})$, respectively, set $(\Theta_j^\intercal X)^n := (\Theta_j^\intercal X_1, \ldots, \Theta_j^\intercal X_n)$ for $j = 1, \ldots, m$, and similarly define $(\Phi_j^\intercal Y)^n$. We consider the following SMI estimator:

$$\widehat{\mathsf{SI}}_{n,m} = \widehat{\mathsf{SI}}_{n,m}(X^n, Y^n, \Theta^m, \Phi^m) := \frac{1}{m} \sum_{i=1}^m \hat{\mathsf{I}}\big((\Theta_i^\intercal X)^n, (\Phi_i^\intercal Y)^n\big). \tag{6}$$

Pseudocode and computational complexity for $\widehat{\mathsf{SI}}_{n,m}$ can be found in Section B of the supplement.

### 3.1.1 Non-asymptotic performance guarantees

We now present convergence guarantees for the estimator (6) over the following class of distributions:

$$\mathcal{F}_{d_x,d_y}(M) := \left\{ P_{X,Y} \in \mathcal{P}(\mathbb{R}^{d_x} \times \mathbb{R}^{d_y}) : \begin{array}{l} \sup_{(\theta,\phi) \in \mathbb{S}^{d_x-1} \times \mathbb{S}^{d_y-1}} \mathsf{I}(\theta^\intercal X; \phi^\intercal Y) \leq M, \\ (\pi^\theta, \pi^\phi)_\sharp P_{X,Y} \in \mathcal{F}, \forall (\theta, \phi) \in \mathbb{S}^{d_x-1} \times \mathbb{S}^{d_y-1} \end{array} \right\},$$

i.e., the class of all $P_{X,Y}$ with bounded MI and projections that belong to the class $\mathcal{F}$ from (5).

**Theorem 1** (Convergence rate). *The following uniform error bound over $\mathcal{F}_{d_x,d_y}(M)$ holds:*

$$\sup_{P_{X,Y} \in \mathcal{F}_{d_x,d_y}(M)} \mathbb{E}\Big[\big|\mathsf{SI}(X;Y) - \widehat{\mathsf{SI}}_{n,m}\big|\Big] \leq \frac{M}{2\sqrt{m}} + \delta(n). \tag{7}$$

See Supplement A.4 for the proof.

**Remark 5** (Instance-dependent bound). *An inspection of the proof of Theorem 1 reveals that (7) can be converted into the instance-dependent bound*

$$\mathbb{E}\big[\big|\mathsf{SI}(X;Y) - \widehat{\mathsf{SI}}_{n,m}\big|\big] \leq \frac{1}{2} \sup_{(\theta,\phi) \in \mathbb{S}^{d_x-1} \times \mathbb{S}^{d_y-1}} \mathsf{I}(\theta^\intercal X; \phi^\intercal Y) \, m^{-1/2} + \delta(n),$$

*so long that $P_{X,Y} \in \mathcal{F}_{d_x,d_y}(M)$ for some $M$.*

Theorem 1 applies to classes of distribution with uniformly bounded per-slice MI. Since this boundedness may be hard to verify in practice, we present a primitive sufficient condition for (7) to hold. Specifically, when $P_{X,Y}$ is log-concave and symmetric, it is enough to require that the canonical correlation coefficient of $(X, Y)$ is bounded. Recall that a probability measure $P \in \mathcal{P}(\mathbb{R}^d)$ is called log-concave if for any compact Borel sets $A$ and $B$ and $0 < \lambda < 1$, we have $P\big(\lambda A + (1 - \lambda)B\big) \geq P(A)^\lambda P(B)^{1-\lambda}$. Let

$$\mathcal{F}_{d_x,d_y}^{(\mathsf{LC})}(M) := \big\{ P_{X,Y} \in \mathcal{P}(\mathbb{R}^{d_x} \times \mathbb{R}^{d_y}) : \ P_{X,Y} \text{ is symmetric and log-concave}, \ \rho_{\mathsf{CCA}}^2(X,Y) \leq M \big\}.$$

The following error bound is proved in Supplement A.5.

**Corollary 1** (Convergence for log-concave class)**.** *The following uniform error bound over* $\mathcal{F}_{d_x,d_y}^{(\mathsf{LC})}(M)$ *holds:*

$$\sup_{P_{X,Y} \in \mathcal{F}_{d_x,d_y}^{(\mathsf{LC})}(M)} \mathbb{E}\Big[\big|\mathsf{SI}(X;Y) - \widehat{\mathsf{SI}}_{n,m}\big|\Big] \leq \left(\frac{1}{2} \log\left(\frac{\pi^2}{8} \frac{1}{1-M}\right) \frac{1}{m}\right)^{-1/2} + \delta(n). \quad (8)$$

### 3.1.2 End-to-end SMI estimation guarantees over Lipschitz balls

To provide a concrete SMI estimator with guarantees, we instantiate the low-dimensional MI estimate via the entropy estimator from [26] for densities in the generalized Lipschitz class.

**Definition 4** (Generalized Lipschitz class)**.** *For* $d \in \mathbb{N}$, $p \in [2, \infty)$, $s > 0$, *and* $L \geq 0$, *let* $\mathsf{Lip}_{s,p,d}(L)$ *be the class of probability density functions* $f : [0,1]^d \to \mathbb{R}$ *with* $\|f\|_{\mathsf{Lip}_{s,p,d}} \leq L$, *where*

$$\|f\|_{\mathsf{Lip}_{s,p,d}} := \|f\|_{p,d} + \sup_{t>0} t^{-s} \sup_{e \in \mathbb{R}^d, \|e\|_2 \leq 1} \big\|\Delta_{te}^{\lceil s \rceil} f\big\|_{p,d} \quad (9)$$

*and* $\Delta_h^r f(x) := \sum_{k=0}^r (-1)^{r-k} \binom{r}{k} f\big(x + \big(k - \frac{r}{2}\big) h\big)$.

We note that the norm of $\Delta_{te}^{\lceil s \rceil} f$ is taken over the whole Euclidean space to enforce a smooth decay of $f$ at the boundary. Consequently, the $\mathsf{Lip}_{s,p,d}(L)$ class includes, e.g., densities whose derivatives up to order $\lceil s \rceil - 1$ all vanish at the boundary, where $s$ is the smoothness parameter.

Differential entropy estimation over $\mathsf{Lip}_{s,p,d}(L)$ was considered in [26], where an optimal estimator based on best polynomial approximation and kernel density estimation techniques was proposed. Adhering to their setup, for $A \sim P_A \in \mathcal{P}([0,1]^d)$ with density $p_A \in \mathsf{Lip}_{s,p,d}(L)$, we denote the aforementioned entropy estimate based on $n$ i.i.d. samples $A^n$ by $\hat{\mathsf{H}}(A^n)$.

The SMI estimator from (6) employs a MI estimate between scalar variables $(A, B)$. Assume their joint density is $p_{A,B} \in \mathsf{Lip}_{s,p,2}(L)$ and let $(A^n, B^n)$ be i.i.d. samples. To estimate $\mathsf{I}(A;B)$, consider

$$\hat{\mathsf{I}}_{\mathsf{Lip}}(A^n; B^n) := \hat{\mathsf{H}}(A^n) + \hat{\mathsf{H}}(B^n) - \hat{\mathsf{H}}(A^n, B^n). \quad (10)$$

Plug (10) into (6) and let $\widehat{\mathsf{SI}}_{n,m}^{(\mathsf{Lip})}$ be the resulting SMI estimate. We next state the effective estimation rate over $\mathcal{F}_{s,p,d_x,d_y}^{(\mathsf{Lip})}(L, M) := \big\{ p_{X,Y} \in \mathsf{Lip}_{s,p,d_x+d_y}(L) : \ \sup_{(\theta,\phi) \in \mathbb{S}^{d_x-1} \times \mathbb{S}^{d_y-1}} \mathsf{I}(\theta^\mathsf{T} X; \phi^\mathsf{T} Y) \leq M \big\}$.

**Corollary 2** (Effective rate)**.** *Let* $d_x, d_y \in \mathbb{N}$, $s \in (0, 2]$, $p \in [2, \infty)$, *and* $L \geq 0$. *The following uniform error bound over* $\mathcal{F}_{s,p,d_x,d_y}^{(\mathsf{Lip})}(L, M)$ *holds:*

$$\sup_{p_{X,Y} \in \mathcal{F}_{s,p,d_x,d_y}^{(\mathsf{Lip})}(L,M)} \mathbb{E}\Big[\big|\mathsf{SI}(X;Y) - \widehat{\mathsf{SI}}_{n,m}^{(\mathsf{Lip})}\big|\Big] \leq \frac{M}{2} m^{-\frac{1}{2}} + C\Big((n \log n)^{-\frac{s}{s+2}} (\log n)^{\left(1 - \frac{2}{p(s+2)}\right)} + n^{-\frac{1}{2}}\Big).$$

*for a constant* $C$ *that depends only on* $d_x$, $d_y$, $p$, $s$, *and* $L$.

The proof of Corollary 2 (Supplement A.6) shows that densities in the generalized Lipschitz class have the property that all their projections are also in that class (with different parameters). We then bound $\delta(n)$ using [26, Theorem 4] to control the error of each differential entropy estimate in (10).

**Remark 6** (SMI versus MI estimation rates)**.** *The SMI estimation rate from Corollary 2 is considerably faster than the* $n^{-1/(d_x+d_y)}$ *rate attainable when estimating classic MI [26]. Our bound shows*

that $n$ and $(d_x, d_y)$ are decoupled in the SMI convergence rate, unlike their interleaved dependence in MI estimation. The ambient dimension still enters the bound via the constant $C$, but its effect is expected to be much milder than in the classic case. As Theorem 1 shows, $(d_x, d_y)$ can only enter via $M$ and $\delta(n)$, both of which correspond to scalar MI terms (namely, the uniform per-sliced MI bound and scalar MI estimation error). This scalability to high dimensions is the expected gain from slicing.

**Remark 7** (Optimal rate for smooth densities). *Restricting attention to densities of maximal smoothness in Corollary 2, i.e., $s = 2$, the resulting rate is $0.5Mm^{-1/2} + Cn^{-1/2}\big(1 + (\log n)^{0.5(1-1/p)}\big)$. Equating the number of MC and data samples, $m$ and $n$, the rate is parametric, up to polylog factors.*

## 3.2 Extracting Sliced Information

We now discuss how SMI can increase via processing. In contrast to the DPI of classic MI, we show that SMI can be grow by extracting linear features of $X$ and $Y$ that are informative of each other.

To illustrate the idea we begin with a simple example. Let $X = (X_1\ X_2) \sim \mathcal{N}(0, I_2)$, $Y = X_1$, and consider $\mathsf{SI}(X; Y)$ ($Y$ is a scalar and it is thus not sliced). For any $\theta = (\theta_1\ \theta_2) \in \mathbb{S}^1$ with $\theta_1 \neq 0$, we have $\mathsf{I}(\theta^\mathsf{T} X; Y) = \mathsf{I}\big(X_1 + (\theta_2/\theta_1)X_2; X_1\big) = \frac{1}{2}\log\big(1 + (\theta_1/\theta_2)^2\big)$, where the last step uses the independence of $X_1$ and $X_2$. Consider the function $g_a : \mathbb{R}^2 \to \mathbb{R}^2$ given by $g_a(x_1, x_2) = (x_1\ ax_2)^\mathsf{T}$, for some $0 < a < 1$. Following the same procedure, we have $\theta^\mathsf{T} g_a(X) = \theta_1 Y + a\theta_2 X_2$ and $\mathsf{I}\big(\theta^\mathsf{T} g_a(X); Y\big) = \frac{1}{2}\log\left(1 + \big(\theta_1/(a\theta_2)\big)^2\right) > \mathsf{I}\big(\theta^\mathsf{T} X; Y\big)$, for almost all $\theta \in \mathbb{S}^1$, and consequently, $\mathsf{SI}(X; Y) < \mathsf{SI}\big(g_a(X); Y\big)$. Generally, this shows that by varying $a$ one can both create and diminish sliced information by processing via $g_a$.

When $a = 0$, we have $\mathsf{SI}(g_a(X); Y) = \infty$, yielding $\mathsf{SI}(g_a(X); Y) = \sup_\theta \mathsf{I}(\theta^\mathsf{T} X; Y)$. Thus, maximizing SMI by varying $a$ extracts the most informative feature $X_1$ and deletes the uninformative feature $X_2$. We next generalize this observation (see Supplement A.7 for the proof).

**Proposition 4.** *Let $(X, Y) \sim \mathcal{P}(\mathbb{R}^{d_x} \times \mathbb{R}^{d_y})$ and consider optimizing the SMI between linear processing of $X$ and $Y$ in the following scenarios:*

1. ***Arbitrary linear processing****: For matrices $\mathrm{A}_x, \mathrm{A}_y$ and vectors $b_x, b_y$ of the appropriate dimension, we have*

$$\sup_{\mathrm{A}_x, \mathrm{A}_y, b_x, b_y} \mathsf{SI}(\mathrm{A}_x X + b_x; \mathrm{A}_y Y + b_y) = \sup_{\mathrm{A}_x, \mathrm{A}_y} \mathsf{SI}(\mathrm{A}_x X; \mathrm{A}_y Y) = \sup_{\theta, \phi} \mathsf{I}(\theta^\mathsf{T} X; \phi^\mathsf{T} Y). \quad (11)$$

   *Furthermore, if $(\theta^\star, \phi^\star) \in \arg\max \mathsf{I}(\theta^\mathsf{T} X; \phi^\mathsf{T} Y)$, then an optimal pair of matrices $\mathrm{A}_x^\star$ and $\mathrm{A}_y^\star$ have $\theta^\star$ or $\phi^\star$, respectively, in their first rows and zeros otherwise.*

2. ***Rank constrained linear processing****: For $d_1, d_2, r \in \mathbb{N}$ and $c > 0$, let $\mathcal{M}_{d_1, d_2}(r, c) := \big\{\mathrm{A} \in \mathbb{R}^{d_1 \times d_2} : \frac{1}{c} \leq \sigma_r(\mathrm{A}) \leq \ldots \leq \sigma_1(\mathrm{A}) \leq c\big\}$, where $\sigma_i(\mathrm{A})$ is the $i$th largest singular value of $\mathrm{A}$. We have*

$$\sup_{\substack{\mathrm{A}_x \in \mathcal{M}_{d_x, d_x}(r_x, c_x), \\ \mathrm{A}_y \in \mathcal{M}_{d_x, d_x}(r_y, c_y)}} \mathsf{SI}(\mathrm{A}_x X; \mathrm{A}_y Y) = \sup_{\substack{\mathrm{B}_x \in \mathcal{M}_{r_x, d_x}(r_x, c_x), \\ \mathrm{B}_y \in \mathcal{M}_{r_y, d_y}(r_y, c_y)}} \mathsf{SI}(\mathrm{B}_x X; \mathrm{B}_y Y), \quad \forall c_x, c_y > 0.$$

   *Furthermore, if $(\mathrm{B}_1^\star, \mathrm{B}_2^\star)$ are maximizers of the RHS, then $\mathrm{B}_1$ (resp., $\mathrm{B}_2$) has the first $r_x$ (resp., $r_y$) rows span the top $r_x$ (resp., $r_y$) scalar MI slicing directions and the remaining rows are zero.*

The proposition suggests that SMI can be used as an objective for extracting informative linear features. The setup in Case 2 precludes reduction to Case 1. Indeed, if eigenvalues can shrink or grow without bound, it is always better to consider a maximizing slice than to average several slices.

**Remark 8** (Processing one variable). *Similar results hold when only one of the arguments ($X$ or $Y$) is processed. In this case, rather than the maximum being a projected MI term, it would be an SMI where the slicing is only in the opposite argument. For example, (11) becomes $\sup_{\mathrm{A}_x, b_x} \mathsf{SI}(\mathrm{A}_x X + b_x; Y) = \sup_{\mathrm{A}_x} \mathsf{SI}(\mathrm{A}_x X; Y) = \sup_\theta \mathsf{I}(\theta^\mathsf{T} X; \Phi^\mathsf{T} Y | \Phi)$, for $\Phi \sim \mathsf{Unif}(\mathbb{S}^{d_y - 1})$ independent of $(X, Y)$.*

Proposition 4 accounts for linear processing but the argument readily extends to nonlinear processing. For simplicity, the following corollary states the result for a shallow (single hidden layer) neural network (see Supplement A.8 for the proof).

**Corollary 3** (Shallow neural network). *Let $(X, Y) \sim P_{X,Y}$. For any scaling matrices $\mathrm{A}_x, \mathrm{A}_y$, weight matrices $\mathrm{W}_x, \mathrm{W}_y$, and bias vectors $b_x, b_y$ of appropriate dimension, define $X_{\mathsf{nn}} := \mathrm{A}_x \sigma(\mathrm{W}_x^\mathsf{T} X + b_x)$, $Y_{\mathsf{nn}} := \mathrm{A}_y \sigma(\mathrm{W}_y^\mathsf{T} Y + b_y)$, where $\sigma$ is a scalar, continuous, and monotonically increasing nonlinearity (e.g. sigmoid, tanh) and the hidden dimension is arbitrary. Then*

$$\sup_{\mathrm{A}_x, \mathrm{A}_y, \mathrm{W}_x, \mathrm{W}_y, b_x, b_y} \mathsf{SI}(X_{\mathsf{nn}}; Y_{\mathsf{nn}}) = \sup_{\theta, \phi, \mathrm{W}_x, \mathrm{W}_y, b_x, b_y} \mathsf{I}\big(\theta^\mathsf{T} \sigma(\mathrm{W}_x^\mathsf{T} X + b_x); \phi^\mathsf{T} \sigma(\mathrm{W}_y^\mathsf{T} Y + b_y)\big).$$

# 4 Empirical Results

## 4.1 Convergence of the SMI estimator

We validate the empirical convergence rates for SMI estimation derived in Section 3.1. Consider densities with smoothness parameter $s = 2$ in the setup of Corollary 2; the expected convergence rate (up to log factors) is $n^{-1/2} + m^{-1/2}$. While the theoretical results use the optimal estimator of [26] to obtain the tightest bounds, in our experiments we implemented the simpler Kozachenko–Leonenko estimator. The justification for doing so comes from [25], who showed that this estimator achieves the same rate (up to log factors) as the optimal one from [26].

Figure 1 shows convergence of the estimated SMI RMSE for the case where $X$ and $Y$ are overlapping subsets of a standard normal random vector $Z \sim \mathcal{N}(0, \mathrm{I}_{15})$. For $d = 3$, we set $X = Z_{[1:3]} := (Z_1\ Z_2\ Z_3)^\mathsf{T}$, $Y = Z_{[2:4]}$ (i.e., 2 coordinates overlap). For $d = 10$, we take $X = Z_{1:10}$, $Y = Z_{5:15}$ (5 coordinates overlap). Convergence is shown when both $n$ and $m$ grow together (i.e., $n = m$), and when one is fixed to a large value and the other varies. The large value is chosen so that the error term corresponding to the fixed parameter is negligible compared to the varying term. For $d = 10$, we also plot results for $m, n$ varying independently. Supplement C provides corresponding MI estimation results.

## 4.2 Independence testing

Proposition 1 states that $\mathsf{SI}(X; Y) = 0$ if and only if $X$ and $Y$ are independent. This implies that, like MI, we may use SMI for independence testing. MI-based independence tests of high-dimensional continuous variables can be burdensome due to slow convergence of MI estimation [26]. We show that, as our theory implies, SMI is a scalable alternative.

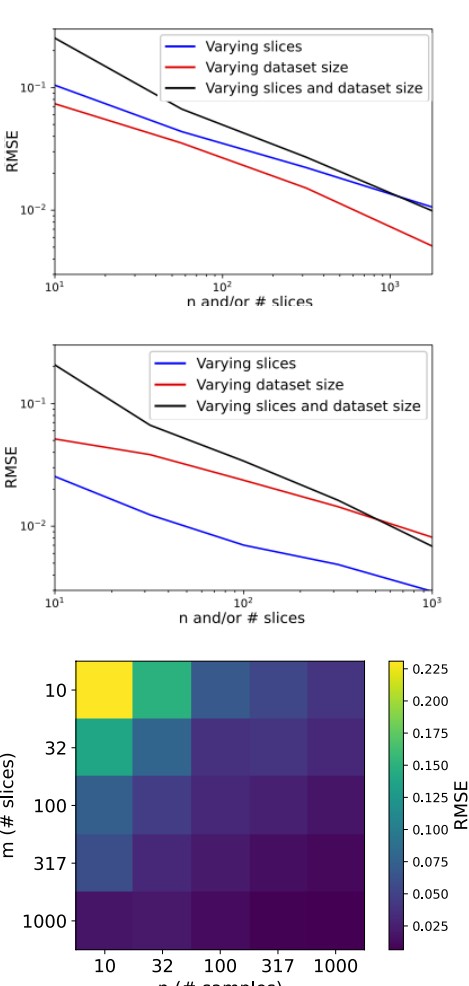

Figure 1: Convergence of the SMI estimator versus the number of data samples $n$ and/or slice samples $m$: (a) $d = 3$, $m_{\text{fixed}} = n_{\text{fixed}} = 10^4$; (b) $d = 10$, $m_{\text{fixed}} = n_{\text{fixed}} = 10^4$; (c) $d = 10$, $m, n$ varied independently.

Figure 2 shows independence testing results for a variety of relationships between $X, Y$ pairs. The figure shows the area under the curve (AUC) of the receiver operating characteristic (ROC) for independence testing via SMI (or MI) thresholding as a function of the number of samples $n$ from the joint distribution.[4] Both the SMI and MI were computed using the Kozachenko–Leonenko estimator [15]; the MC step for SMI estimation (see (6)) uses 1000 random slices, and the AUC ROC curves are computed from 100 random trials. The joint distribution of $(X, Y)$ in each case of Figure 2 is:

---

[4]For every $n$, we generate 50 datasets comprising $n$ positive samples (i.e., drawn from the joint distribution) and 50 more dataset of negative samples in each setting. SMI and MI are then estimated over each dataset, the ROC curve is found, and the area under it computed. The ROC curve plots test performance (precision and recall) as the threshold is varied over all possible values. The AUC ROC quantifies the test's discriminative ability: an omniscient classifier has AUC ROC 1, while random tests have AUC ROC 0.5.

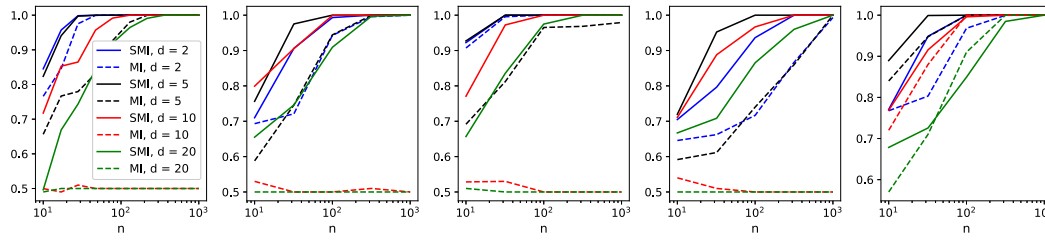

(a) $Y$ encodes single $X$ feature (lin)  (b) $Y$ encodes single $X$ feature (sin)  (c) $Y$ encodes two features of $X$  (d) Low rank common signal  (e) Independent coordinates

Figure 2: Statistical efficiency of independence testing with dimension. The plot show the area under the ROC curve (AUC ROC) as a function of samples $n$ for several dimensions $d$. The test is based on thresholding SMI and MI. Details for each scenario are in the text.

(a) **One feature (linear):** $X, Z \sim \mathcal{N}(0, \mathrm{I}_d)$ i.i.d. and $Y = \frac{1}{\sqrt{2}}\big(\frac{1}{\sqrt{d}}(\mathbf{1}^\intercal X)\mathbf{1} + Z\big)$, where $\mathbf{1} := (1 \ldots 1)^\intercal \in \mathbb{R}^d$.

(b) **One feature (sinusoid):** $X, Z \sim \mathcal{N}(0, \mathrm{I}_d)$ i.i.d. and $Y = \frac{1}{\sqrt{2}}\big(\frac{1}{\sqrt{d}}\sin(\mathbf{1}^\intercal X)\mathbf{1} + Z\big)$.

(c) **Two features:** $X, Z \sim \mathcal{N}(0, \mathrm{I}_d)$ i.i.d. and $Y_i = \frac{1}{\sqrt{2}} \begin{cases} \frac{1}{d}(\mathbf{1}_{\lfloor d/2 \rfloor} 0 \ldots 0)^\intercal X + Z_i, & i \leq \frac{d}{2} \\ \frac{1}{d}(0 \ldots 0 \mathbf{1}_{\lceil d/2 \rceil})^\intercal X + Z_i, & i > \frac{d}{2}. \end{cases}$

(d) **Low rank common signal:** $Z_1, Z_2 \sim \mathcal{N}(0, \mathrm{I}_d)$ and $V \sim \mathcal{N}(0, \mathrm{I}_2)$ are independent; $X = \mathrm{P}_1 V + Z_1$ and $Y = \mathrm{P}_2 V + Z_2$, where $\mathrm{P}_1, \mathrm{P}_2 \in \mathbb{R}^{d \times 2}$ are projection matrices (realized at the beginning of each iteration by drawing i.i.d. standard normal entries).

(e) **Independent coordinates:** $X, Z \sim \mathcal{N}(0, \mathrm{I}_d)$ i.i.d. and $Y = \frac{1}{\sqrt{2}}(X + Z)$.

Note that in all the cases with underlying lower-dimensional structure, SMI scales well with dimension while MI does not; in the independent case of subfigure (d), both perform similarly and rather well. While the SMI is always on par or better than MI in these experiments, the results suggest that SMI performs best in structured (specifically, low rank) settings. This is because in these settings the MI term associated with random $(\Theta, \Phi)$ slices has lower variance. This is not the case for the unstructured setting of Figure 2(d). There, when dimension is high, random slices carry relatively little information compared to the maximum MI over slices (which attained between the $i$th coordinates of $X$ and $Y$ for any $i$). Since SMI is an average quantity, in this case it offers little gain over classic MI.

### 4.3 Feature extraction

In the above, we focused on a nonparametric estimator for which we derived tight bounds. In practice, applying neural estimators (à la MINE [29]) is more compatible with modern optimizers. The SMI neural estimator (S-MINE) relies on the variational representation from Proposition 3. Given a sample set $(X_1, Y_1), \ldots, (X_n, Y_n)$ i.i.d. from $P_{X,Y}$, we further sample $n$ i.i.d. copies of $(\Theta, \Phi) \sim \mathsf{Unif}(\mathbb{S}^{d_x - 1}) \times \mathsf{Unif}(\mathbb{S}^{d_y - 1})$. The negative samples $(\tilde{X}_1, \tilde{Y}_1), \ldots, (\tilde{X}_n, \tilde{Y}_n)$ are obtained by permuting the order of, e.g., the $Y$ samples. Parametrizing the potential $g$ in (3) by a neural network $g_\xi$ with parameters $\xi \in \Xi$, we obtain the following empirical objective

$$\sup_{\xi \in \Xi} \frac{1}{n} \sum_{i=1}^n g_\xi(\Theta_i, \Phi_i, \Theta_i^\intercal X_i, \Phi_i^\intercal Y_i) - \log\left(\frac{1}{n} \sum_{i=1}^n e^{g_\xi(\Theta_i, \Phi_i, \Theta_i^\intercal \tilde{X}_i, \Phi_i^\intercal \tilde{Y}_i)}\right).$$

This provides an estimate (from below) of $\mathsf{SI}(X; Y)$. We leave a full theoretical and empirical exploration of its performance for future work, and here only provide two proof-of-concept experiments.

The formulation of SMI as an optimization of a loss whose gradients are readily evaluated lends itself to the SMI-extraction formulations of Proposition 4, i.e., we can simultaneously optimize $\mathrm{A}_x$, $\mathrm{A}_y$, and $\xi$ end-to-end. Figure 3 shows results for $X, Z \sim \mathcal{N}(0, \mathrm{I}_{10})$, $Y = e_1^\intercal X + Z$, where $e_1 = (1\ 0 \ldots 0)^\intercal$, with $g_\xi$ in S-MINE realized as a two-layer fully connected neural network with 100 hidden units for. An $(\mathrm{A}_x^\star, \mathrm{A}_y^\star)$ with rows converging to $e_1$ are recovered. Note that while Proposition 4 identifies an optimal solution where only the first row is nonzero, this will have the same SMI as when all rows equal that first row. The latter solution is found since the gradients do not favor one row over another.

We next combine S-MINE with independence testing, looking to maximize the SMI using transformations $AX, AY$, where $X, Y$ are samples from a random MNIST class (either 0 or 1) and $A \in \mathbb{R}^{10 \times d}$. Specifically, we choose a class $C \sim \text{Ber}(0.5)$ and then sample $X$ and $Y$ uniformly from that class' training dataset. In this setup, $X, Y$ share up to 1 bit of information, i.e. $\mathsf{I}(X;Y) \leq \mathsf{H}(C) = \log 2$. This suggests that maximizing the SMI between $AX$ and $AY$ will find an $A$ that extracts information about $C$, revealing whether $X$ and $Y$ are in the same class. Optimizing $A$ yields an estimated SMI of 0.680 bits (compare this to, e.g., 0.0752 SMI achieved by a matrix $A$ with i.i.d. standard normal entries). To confirm $A$ is not being overfit, we also optimized the SMI over $A$ when $X, Y$ are drawn independently, i.e., no longer sharing a class, yielding 0.0289 estimated SMI (the ground truth is 0 here). These results indicate that (a) an

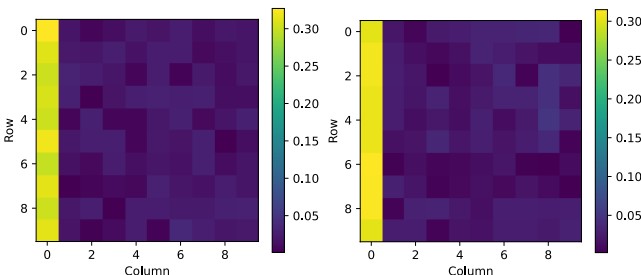

Figure 3: $(A_x^\star, A_y^\star)$ optimizers for (11) over a Gaussian dataset using S-MINE: (a) matrix $A_x^\star$; (b) matrix $A_y^\star$. Both have correctly recovered the unit vector $e_1$ as their rows.

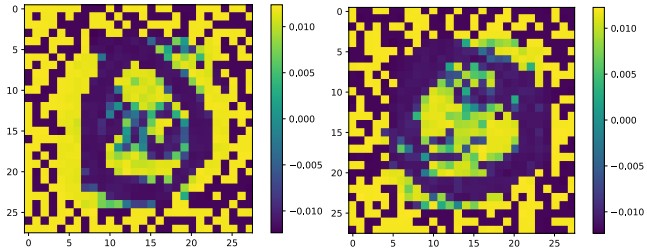

Figure 4: Solution for optimizing $A$-transformed SMI of the 0-1 MNIST setup using S-MINE. Rows 0 and 1 of $A$ are shown in (a) and (b), respectively.

SMI-based independence test would be successful at detecting dependence between $X, Y$ and (b) optimizing $A$ not only succeeds at significantly (almost 10x) increasing the SMI, but also comes relatively close to the true MI of 1 bit. Heatmaps of two rows of $A$ rearranged into the MNIST image dimensions are shown in Figure 4. Observe the $A_{i:}$ visually correspond to the numeral 0, which, from a matched filter perspective, yields an $AX$ (resp., $AY$) that is informative of whether $X$ (resp., $Y$) is in class zero or not. This, in turn, conveys information of whether $X, Y$ share a class (since 0 and 1 are the only options), as desired.

## 5  Summary and Concluding Remarks

Motivated to address the computational and statistical unscalability of MI to high dimensions, this paper proposed an alternative notion of informativeness dubbed sliced mutual information (SMI). SMI projects high-dimensional random variables to scalars and averages over random projections. We showed that SMI shares many of the structural properties of classic MI, while enjoying efficient empirical estimation. We also showed that, as opposed to classic MI, SMI can be increase by processing the variables. This observation was quantified for linear and nonlinear SMI-based feature extractors. Experiments validating the theoretical study were provided, demonstrating dimension free empirical convergence rates, statistical efficiency for independence testing, and feature extraction examples.

Our results pose SMI as a favorable figure of merit for information quantification between high-dimensional random variables. We expect it to turn useful for a variety of applications in inference and machine learning, although a large-scale empirical exploration is beyond the scope of this paper. In particular, SMI seems well adapted for representation learning via the (sliced) InfoMax principle [30, 31], and we plan to test this hypothesis in future work. On the theoretical side, appealing future directions are abundant, including convergence guarantees for the neural SMI estimator used in Section 4.3, operational channel/source coding settings for which SMI characterizes the information-theoretic fundamental limit, and a statistical analysis of SMI-based independence testing.

## Acknowledgements

Z. Goldfeld is supported by the NSF CRII Grant CCF-1947801, the NSF CAREER Award under Grant CCF-2046018, and the 2020 IBM Academic Award.

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
