# Supplementary Materials for:
# Sliced Mutual Information: A Scalable Measure of Statistical Dependence

**Ziv Goldfeld**
Cornell University
goldfeld@cornell.edu

**Kristjan Greenewald**
MIT-IBM Watson AI Lab
kristjan.h.greenewald@ibm.com

## A   Proofs

### A.1   Proof of Proposition 1

**Proof of 1.**   $\mathsf{SI}(X;Y) \geq 0$ is trivial by non-negativity of conditional MI. For the equality to zero case, recall that $X$ and $Y$ are independent if and only if (iff) their joint characteristic function $\varphi_{X,Y}(t,s) := \mathbb{E}\left[e^{itX+isY}\right]$ decomposes into a product, i.e.,

$$\varphi_{X,Y}(t,s) = \varphi_X(t)\varphi_Y(s) = \mathbb{E}\left[e^{itX}\right]\mathbb{E}\left[e^{isY}\right], \quad \forall t,s \in \mathbb{R}.$$

Also recall that independence is equivalent to zero classic mutual information. Denote $X_\theta := \theta^\mathsf{T} X$ and $Y_\phi := \phi^\mathsf{T} Y$ and observe that $\mathsf{SI}(X;Y) = 0$ is equivalent to

$$\oint_{\mathbb{S}^{d_x-1}} \oint_{\mathbb{S}^{d_y-1}} \mathsf{I}(X_\theta;Y_\phi)\mathrm{d}\theta\mathrm{d}\phi = 0. \tag{12}$$

Indeed, as $\mathsf{I}(X_\theta;Y_\phi) \geq 0$, for any $(\theta,\phi) \in \mathbb{S}^{d_x-1} \times \mathbb{S}^{d_y-1}$, (12) holds iff

$$\varphi_{X_\theta,Y_\phi}(t,s) = \varphi_{X_\theta}(t)\varphi_{Y_\phi}(s), \quad \forall t,s \in \mathbb{R},$$

but this is the same as

$$\varphi_{X,Y}(t\theta,s\phi) = \varphi_X(t\theta)\varphi_Y(s\phi), \quad \forall t,s \in \mathbb{R},\ \theta \in \mathbb{S}^{d_x-1},\ \phi \in \mathbb{S}^{d_y-1}.$$

Changing variables $t' = t\theta$ and $s' = s\phi$, the last equality holds iff

$$\varphi_{X,Y}(t',s') = \varphi_X(t')\varphi_Y(s'), \quad \forall t' \in \mathbb{R}^{d_x},\ s' \in \mathbb{R}^{d_y},$$

which means $X$ and $Y$ are independent.

**Proof of 2.**   Since SMI is an average of projected MI terms we immediately have

$$\inf_{\theta\in\mathbb{S}^{d_x-1},\phi\in\mathbb{S}^{d_y-1}} \mathsf{I}(\theta^\mathsf{T} X;\phi^\mathsf{T} Y) \leq \mathsf{SI}(X;Y) \leq \sup_{\theta\in\mathbb{S}^{d_x-1},\phi\in\mathbb{S}^{d_y-1}} \mathsf{I}(\theta^\mathsf{T} X;\phi^\mathsf{T} Y).$$

By the DPI for classic MI we further upper bound the right-hand side (RHS) by $\mathsf{I}(X;Y)$.

We further note that the infimum in the lower bound is always attained, as is thus a minimum. This is because for any $(\theta_n,\phi_n),(\theta,\phi) \in \mathbb{S}^{d_x-1} \times \mathbb{S}^{d_y-1}$ with $\theta_n \to \theta$ and $\phi_n \to \phi$, we have that $(\theta_n^\mathsf{T} X, \phi_n^\mathsf{T} Y)$ converge to $(\theta^\mathsf{T} X, \phi^\mathsf{T} Y)$ almost surely (in fact, surely) and therefore in distribution. Since MI is weakly lower semicontinuous, it attains a minimum on the compact set $\mathbb{S}^{d_x-1} \times \mathbb{S}^{d_y-1}$. To attain the supremum one must impose additional regularity on the Lebesgue density of $P_{X,Y}$ to ensure that MI is continuous in the weak topology; see, e.g., [32, Theorem 1].

**Proof of 3.** This follows because conditional mutual information can be expressed as

$$\mathsf{I}(X; Y | Z) = \mathbb{E}_Z\Big[\mathsf{D}_{\mathsf{KL}}\big(P_{X,Y|Z}(\cdot | Z) \big\| P_{X|Z}(\cdot | Z) \otimes P_{Y|Z}(\cdot | Z)\big)\Big],$$

and because the joint distribution of $(\Theta^{\mathsf{T}} X, \Phi^{\mathsf{T}} Y)$ given $\{\Theta = \theta, \Phi = \phi\}$ is $(\pi^\theta, \pi^\phi)_\sharp P_{X,Y}$, while the corresponding conditional marginals are $\pi^\theta_\sharp P_X$ and $\pi^\phi_\sharp P_Y$, respectively.

**Proof of 4.** We only prove the small chain rule; generalizing to $n$ variables is straightforward. Consider:

$$\begin{aligned}
\mathsf{SI}(X, Y | Z) &= \mathsf{I}(\Theta^{\mathsf{T}} X, \Phi^{\mathsf{T}} Y; \Psi^{\mathsf{T}} Z | \Theta, \Phi, \Psi) \\
&= \mathsf{I}(\Theta^{\mathsf{T}} X; \Psi^{\mathsf{T}} Z | \Theta, \Phi, \Psi) + \mathsf{I}(\Phi^{\mathsf{T}} Y; \Psi^{\mathsf{T}} Z | \Theta, \Phi, \Psi, \Theta^{\mathsf{T}} X),
\end{aligned}$$

where the last equality is the regular chain rule. Since $(X, Z, \Theta, \Psi)$ are independent of $\Phi$, we have

$$\mathsf{I}(\Theta^{\mathsf{T}} X; \Psi^{\mathsf{T}} Z | \Theta, \Phi, \Psi) = \mathsf{I}(\Theta^{\mathsf{T}} X; \Psi^{\mathsf{T}} Z | \Theta, \Psi) = \mathsf{SI}(X; Z).$$

We conclude the proof by noting that

$$\begin{aligned}
\mathsf{I}(\Phi^{\mathsf{T}} Y; \Psi^{\mathsf{T}} Z | \Theta, \Phi, \Psi, \Theta^{\mathsf{T}} X) &= \frac{1}{S_{d_x-1}} \oint_{\mathbb{S}^{d_x-1}} \mathsf{I}(\Phi^{\mathsf{T}} Y; \Psi^{\mathsf{T}} Z | \Theta = \theta, \Phi, \Psi, \theta^{\mathsf{T}} X) \mathrm{d}\theta \\
&= \frac{1}{S_{d_x-1}} \oint_{\mathbb{S}^{d_x-1}} \mathsf{I}(\Phi^{\mathsf{T}} Y; \Psi^{\mathsf{T}} Z | \Phi, \Psi, \theta^{\mathsf{T}} X) \mathrm{d}\theta \\
&= \mathsf{SI}(Y; Z | X),
\end{aligned}$$

where the penultimate equality is because $(X, Y, Z, \Phi, \Psi)$ are independent of $\Theta$.

**Proof of 5.** By Definition 2, we have

$$\mathsf{SI}(X_1, \ldots, X_n; Y_1, \ldots, Y_n) = \mathsf{I}(\Theta_1^{\mathsf{T}} X_1, \ldots, \Theta_n^{\mathsf{T}} X_n; \Phi_1^{\mathsf{T}} Y_1, \ldots, \Phi_n^{\mathsf{T}} Y_n | \Theta_1, \ldots, \Theta_n, \Phi_1, \ldots, \Phi_n),$$

where the $\Theta_i$, $\Phi_i$ are all independent and uniform on their respective spheres. Now by mutual independence of the $\Theta_i$, $\Phi_i$ and $(X_i, Y_i)$ across $i$,

$$\begin{aligned}
\mathsf{I}(\Theta_1^{\mathsf{T}} X_1, \ldots, \Theta_n^{\mathsf{T}} X_n; \Phi_1^{\mathsf{T}} Y_1, \ldots, \Phi_n^{\mathsf{T}} Y_n | \Theta_1, \ldots, \Theta_n, \Phi_1, \ldots, \Phi_n) &= \sum_{i=1}^n \mathsf{I}(\Theta_i^{\mathsf{T}} X_i; \Phi_i^{\mathsf{T}} Y_i | \Theta_i, \Phi_i) \\
&= \sum_{i=1}^n \mathsf{SI}(X_i; Y_i).
\end{aligned}$$

This concludes the proof. $\qquad\square$

## A.2 Maximum Sliced Entropy and Proof of Proposition 2

In this section we prove the extended claim stated next, which includes Proposition 2 as the first item.

**Proposition 5** (Max sliced entropy)**.** *The following max sliced differential entropy statements hold.*

1. ***Mean and covariance:*** *Let* $\mathcal{P}_1(\mu, \Sigma) := \big\{P \in \mathcal{P}(\mathbb{R}^d) : \operatorname{supp}(P) = \mathbb{R}^d, \mathbb{E}_P[X] = \mu, \mathbb{E}\big[(X - \mu)(X - \mu)^{\mathsf{T}}\big] = \Sigma\big\}$ *be the class of probability measures supported on* $\mathbb{R}^d$ *with fixed mean and covariance. Then*

$$\arg\max_{P \in \mathcal{P}_1(\mu, \Sigma)} \mathsf{SH}(P) = \mathcal{N}(\mu, \Sigma),$$

*i.e. the normal distribution maximizes sliced entropy inside* $\mathcal{P}_1(\mu, \Sigma)$.

2. ***Support contained in a ball:*** *Let* $\mathcal{P}_2(c, r) := \big\{P \in \mathcal{P}(\mathbb{R}^d) : \operatorname{supp}(P) \subseteq \mathbb{B}_d(c, r)\big\}$ *be the class of probability measures supported inside a* $d$-*dimensional ball centered at* $c \in \mathbb{R}^d$ *of radius* $r > 0$ *(denoted by* $\mathbb{B}_d(c, r)$*). Then*

$$\arg\max_{P \in \mathcal{P}_2(c, r)} \mathsf{SH}(P) = \mathsf{Unif}\big(\mathbb{S}^{d-1}(c, r)\big),$$

*i.e. the uniform distribution on the surface of* $\mathbb{B}_d(c, r)$ *maximizes sliced entropy inside* $\mathcal{P}_2(c, r)$.

3. ***Expected absolute deviation:*** *Let* $\mathcal{P}_3(\mu, a) := \big\{ P \in \mathcal{P}(\mathbb{R}^d) : \mathrm{supp}(P) = \mathbb{R}^d, \, \mathbb{E}_P[X] = \mu, \, \mathbb{E}_P|\theta^T(X - \mu)| = a, \, \forall \theta \in \mathbb{S}^{d-1} \big\}$ *be the class of probability measures supported on $\mathbb{R}^d$ with fixed mean and expected absolute deviation of the slice marginals from their mean. Then the sliced entropy inside $\mathcal{P}_3$ is maximized by a $d$-dimensional symmetric multivariate Laplace distribution* [28] *with characteristic function*

$$\Phi(t; \mu, b) = \frac{e^{i\mu^\mathsf{T} t}}{1 + \frac{1}{2} b t^\mathsf{T} t}.$$

*for some $b$ depending on $a$.*

The interpretation of the $\mathbb{E}_P|\theta^T(X - \mu)| = a, \, \forall \theta \in \mathbb{S}^{d-1}$ constraint in 3. is as follows. Note that if the constraint were only for $\theta$s in the cardinal directions (rather than for all $\theta \in \mathbb{S}^{d-1}$), the constraint could be satisfied be the product of i.i.d. Laplace distributions. Unfortunately, the product of Laplace distributions is not a spherical distribution, so the condition would not be satisfied in general for non-cardinal $\theta$. To extend to all $\theta$ on the sphere, it is necessary to find some distribution that is spherical but still has Laplace marginals, in other words, a collection of identically distributed Laplace r.v.s that are coupled such that the joint density is spherical. The Symmetric Multivariate Laplace distribution is exactly this distribution.

*Proof.* For any $P \in \mathcal{P}(\mathbb{R}^d)$ and $\theta \in \mathbb{S}^{d-1}$, denote the distribution of the corresponding projection by $P_\theta := \pi_\sharp^\theta P$. For $X \sim P$, we interchangeably write $\mathsf{H}(X)$ and $\mathsf{H}(P)$ for entropy (similarly, for sliced entropy), and thus express sliced entropy as

$$\mathsf{SH}(P) = \frac{1}{S_{d-1}} \oint_{\mathbb{S}^{d-1}} \mathsf{H}(P_\theta) \mathrm{d}\theta.$$

**Proof of 1.** Note that for any $P \in \mathcal{P}_1(\mu, \Sigma)$ and $\theta \in \mathbb{S}^{d-1}$, the mean and covariance of $P_\theta$ is $\theta^\mathsf{T} \mu$ and $\theta^\mathsf{T} \Sigma \theta$, respectively. Since the Gaussian distribution maximizes classic entropy over scalar distribution supported $\mathbb{R}$ with a fixed (mean and) variance, we have $\mathsf{H}(P_\theta) \leq \mathsf{H}\big(\mathcal{N}(\theta^\mathsf{T} \mu, \theta^\mathsf{T} \Sigma \theta)\big) = \frac{1}{2} \log(2\pi e \theta^\mathsf{T} \Sigma \theta)$ for any $\theta \in \mathbb{S}^{d-1}$. Consequently,

$$\mathsf{SH}(P) \leq \frac{1}{S_{d-1}} \oint_{\mathbb{S}^{d-1}} \frac{1}{2} \log(2\pi e \theta^\mathsf{T} \Sigma \theta) \mathrm{d}\theta, \quad \forall P \in \mathcal{P}_1(\mu, \Sigma). \tag{13}$$

Take $P^\star = \mathcal{N}(\mu, \Sigma) \in \mathcal{P}(\mu, \Sigma)$ and observe that for any $\theta \in \mathbb{S}^{d-1}$, we have $P_\theta^\star = \mathcal{N}(\theta^\mathsf{T} \mu, \theta^\mathsf{T} \Sigma \theta)$. Therefore $\mathsf{SH}(P^\star)$ achieves the upper bound in (13) and is the maximum sliced entropy distribution over $\mathcal{P}_1(\mu, \Sigma)$.

**Proof of 2.** We first show that a maximum entropy distributions over $\mathcal{P}_2(c, r)$ must be rationally invariant and simultaneously maximize the differential entropy associated with each slice. For $X \sim P \in \mathcal{P}(\mathbb{R}^d)$ and an orthogonal matrix $\mathrm{U} \in \mathbb{R}^{d \times d}$, denote (with some abuse of notation) the distribution of $\mathrm{U}X$ by $\mathrm{U}_\sharp P$. Since the support constraint and the definition of sliced entropy are rotationally symmetric, if $P \in \mathcal{P}_2(c, r)$ is a maximum sliced entropy distribution, then so is $\mathrm{U}_\sharp P$, for any $\mathrm{U}$ orthogonal.

Assume $P \in \mathcal{P}_2(c, r)$ maximizes sliced entropy. For any orthogonal $\mathrm{U} \in \mathbb{R}^{d \times d}$ define $\mathcal{A}_\mathrm{U} \subseteq \mathbb{S}^{d-1}$ as the set of $\theta$ vectors for which the distribution of $\theta^\mathsf{T} X$ and $\theta^\mathsf{T} \mathrm{U}X$ are different. Note that if $P$ maximizes $\mathsf{SH}$ then the measure of $\mathcal{A}_\mathrm{U}$ must be zero. Indeed, if this is not the case, consider the mixture distribution $X^\lambda \sim P^\lambda := \lambda P + (1 - \lambda)\mathrm{U}_\sharp P$, and note that by convexity of entropy

$$\mathsf{H}(\theta^\mathsf{T} X^\lambda) > \lambda \mathsf{H}(\theta^\mathsf{T} X) + (1 - \lambda)\mathsf{H}(\theta^\mathsf{T} UX), \qquad \forall \lambda \in (0, 1), \, \theta \in \mathcal{A}_\mathrm{U}.$$

Now, if $\mathcal{A}_\mathrm{U}$ has positive measure, by the definition of sliced entropy we get

$$\mathsf{SH}(X^\lambda) > \frac{1}{S^{d-1}} \oint_{\mathbb{S}^{d-1}} \big(\lambda \mathsf{H}(\theta^\mathsf{T} X) + (1 - \lambda)\mathsf{H}(\theta^\mathsf{T} UX)\big)\mathrm{d}\theta = \lambda \mathsf{SH}(X) + (1 - \lambda)\mathsf{SH}(\mathrm{U}X) = \mathsf{SH}(X),$$

violating the assumption that $X \sim P$ is a maximum sliced entropy distribution. Hence $X \sim P$ is rotationally invariant and has $\mathsf{H}(\theta^\mathsf{T} X)$ invariant with $\theta$, as claimed.

In what follows, we set $c = 0$, the general case is recovered by the translation invariance of entropy. For $d = 3$, by Archimedes' Hat Box Theorem, the projection of the distribution $\mathsf{Unif}(\mathbb{S}^2(0, r))$

onto any $\theta$ yields $\theta^{\mathsf{T}} X \sim \mathsf{Unif}\big([-r,r]\big)$, the entropy-maximizing distribution for the slice. Thus, $P = \mathsf{Unif}(\mathbb{S}^2(0,r))$ maximizes $\mathsf{SH}$ for $d = 3$.

For dimensions $d > 3$, by symmetry we may consider $\theta$ of the form $(\theta_1\ \theta_2\ \theta_3\ 0\ \ldots 0)^{\mathsf{T}}$. Let $X \sim P$ for some rotationally-symmetric distribution $P$. Observe that

$$\theta^T X = (\theta_1\ \theta_2\ \theta_3)(X_1\ X_2\ X_3)^{\mathsf{T}} = (\theta_1\ \theta_2\ \theta_3)\|(X_1\ X_2\ X_3)\|_2 \left( \frac{(X_1\ X_2\ X_3)^{\mathsf{T}}}{\|(X_1\ X_2\ X_3)\|_2} \right).$$

Define $R = \|(X_1\ X_2\ X_3)\|_2$, $\bar{\theta} = (\theta_1\ \theta_2\ \theta_3)^{\mathsf{T}}$, and $\bar{X} = \frac{(X_1\ X_2\ X_3)^{\mathsf{T}}}{\|(X_1\ X_2\ X_3)\|_2}$. By the spherical symmetry of $P$, we have that $\bar{X} \sim \mathsf{Unif}(\mathbb{S}^2(0,1))$ and is independent of $R$. Let $\rho$ be the probability distribution of $R$, and recall that $\mathrm{supp}(\rho) = [0, r]$.

For any fixed $\bar{\theta}$ and $R = r$, by Archimedes' Hat Box Theorem, $r\bar{\theta}^T \bar{X} \sim \mathsf{Unif}\big([-r,r]\big)$. By independence, the density $g$ of $R\bar{\theta}^T \bar{X}$ is then

$$g(t) = \int_0^1 \frac{1}{2\alpha} \mathbb{1}_{\{|t| \leq \alpha\}} d\rho(\alpha), \quad t \in [-r, r],$$

where $\mathbb{1}_A$ is the indicator of $A$. Observe that $g$ is symmetric about 0 and is monotonically nonincreasing away from 0.

We next show that transporting mass in $\rho$ to larger radii values cannot decrease entropy. Let $\epsilon > 0$ and consider moving mass $\epsilon$ in $\rho$ from location $\alpha$ to $\alpha' > \alpha$, changing $g$ to $g'$. Doing so decreases $g$ by $\epsilon\big(1/(2\alpha) - 1/(2\alpha')\big)$ on the interval $t \in (-\alpha, \alpha)$, and increases it by $\epsilon/(2\alpha')$ on the intervals $t \in [-\alpha', -\alpha) \cup (\alpha, \alpha']$. Furthermore, both $g$ and $g'$ monotonically nonincrease away from 0. At $t = \alpha, -\alpha$, set $g = g'$. The corresponding change in entropy is

$$\mathsf{H}(g') - \mathsf{H}(g) = \int g \log g - g' \log g'\, dt$$

$$= 2 \int_\alpha^{\alpha'} [g \log g - g' \log g']dt + 2 \int_0^\alpha [g \log g - g' \log g']dt \qquad (14)$$

We bound these terms separately. Since $g, g'$ are both monotonically non-increasing away from 0,

$$\int_\alpha^{\alpha'} [g \log g - g' \log g']dt \geq \int_\alpha^{\alpha'} \left[ g \log g - g' \left( \log g + \frac{g' - g}{g} \right) \right] dt$$

$$= \int_\alpha^{\alpha'} \left[ (g - g') \left( \log g + \frac{g'}{g} \right) \right] dt$$

$$= -\frac{\epsilon}{2\alpha'} \int_\alpha^{\alpha'} \left[ \log g + \frac{g'}{g} \right] dt$$

$$\geq -\frac{\epsilon}{2\alpha'}(\alpha' - \alpha) \left[ \log g(\alpha) + \frac{g'(\alpha)}{g(\alpha)} \right]$$

$$= -\frac{\epsilon}{2\alpha'}(\alpha' - \alpha) \big[ \log g(\alpha) + 1 \big] \qquad (15)$$

where we have used the concavity of $\log$ to upper bound $\log g' \leq \log g + (g' - g)/g$. Similarly, we have

$$\int_0^\alpha [g \log g - g' \log g']dt \geq \int_0^\alpha \left[ g \log g - g' \left( \log g + \frac{g' - g}{g} \right) \right] dt$$

$$= \int_0^\alpha \left[ (g - g') \left( \log g + \frac{g'}{g} \right) \right] dt$$

$$= \epsilon \left( \frac{1}{2\alpha} - \frac{1}{2\alpha'} \right) \int_0^\alpha \left[ \log g + \frac{g'}{g} \right] dt$$

$$\geq \epsilon \left( \frac{1}{2\alpha} - \frac{1}{2\alpha'} \right) \alpha \left[ \log g(\alpha) + \frac{g'(\alpha)}{g(\alpha)} \right]$$

$$= \epsilon \left( \frac{1}{2\alpha} - \frac{1}{2\alpha'} \right) \alpha \big[ \log g(\alpha) + 1 \big] \tag{16}$$

Substituting (15) and (16) into (14) yields

$$\mathsf{H}(g') - \mathsf{H}(g) \geq 2 \left[ \epsilon \alpha \left( \frac{1}{2\alpha} - \frac{1}{2\alpha'} \right) - \frac{\epsilon}{2\alpha'} (\alpha' - \alpha) \right] \big[ \log g(\alpha) + 1 \big] = 0.$$

Thus, entropy cannot decrease by moving the mass in $\rho$ to larger $R$ values. Note that for any spherically symmetric $X \sim P$ supported in $\mathbb{S}^{d-1}(0, r)$, the transformation $X' \leftarrow r\frac{X}{\|X\|_2}$ yields $R' = \|(X'_1 \, X'_2 \, X'_3)\|_2 = \big\| \frac{r}{\|X\|_2}(X_1 \, X_2 \, X_3) \big\|_2 = \frac{r}{\|X\|_2} R$, i.e. since $\|X\|_2 \leq r$ the transformation uniformly increases $R$ (and thus $\mathsf{H}(g)$), with no change to the distribution of $\bar{X}$. Therefore, $P = \mathsf{Unif}(\mathbb{S}^{d-1}(0, r))$ is the maximum sliced-entropy distribution.

**Proof of 3.** Similar to the Gaussian case of Claim 1, we use the fact that the maximum entropy distribution satisfying $\mathbb{E}|X - \mu| = a$ is the (univariate) Laplace distribution. To maximize the sliced entropy, we thus seek a distribution $P$ that results in each $\theta^T X$ having the same Laplace distribution. Since linear projections of the isotropic Symmetric Multivariate Laplace distribution [28] are all univariate Laplace distributions with the same parameter, this is a maximum sliced entropy distribution for the class. Unfortunately we could not find the exact parameter conversion ($b$ required to achieve $a$) in the literature.

$\square$

## A.3   Proof of Proposition 3

Denote $X_\Theta := \Theta^\intercal X$ and $X_\Phi := \Phi^\intercal X$ and observe that $P_{X_\Theta, Y_\Phi | \Theta, \Phi}(\cdot, \cdot | \theta, \phi) = (\pi^\theta, \pi^\phi)_\sharp P_{X,Y}$. Consider the following two joint distribution:

$$P_{\Theta, \Phi, X_\Theta, Y_\Phi} = P_{\Theta, \Phi} \times P_{X_\Theta, Y_\Phi | \Theta, \Phi}$$
$$Q_{\Theta, \Phi, X_\Theta, Y_\Phi} = P_{\Theta, \Phi} \times P_{X_\Theta | \Theta} \times P_{Y_\Phi | \Phi},$$

where $P_{\Theta, \Phi} = \mathsf{Unif}(\mathbb{S}^{d_x - 1}) \times \mathsf{Unif}(\mathbb{S}^{d_y - 1})$, while $P_{X_\Theta | \Theta}$ and $P_{Y_\Phi | \Phi}$ are the conditional marginals of $P_{X_\Theta, Y_\Phi | \Theta, \Phi}$. By Claim 3 from Proposition 1, we have

$$\mathsf{SI}(X; Y) = \mathsf{D}_{\mathsf{KL}}\big( P_{X_\Theta, Y_\Phi | \Theta, \Phi} \big\| P_{X_\Theta | \Theta} \otimes P_{Y_\Phi | \Phi} \big| P_{\Theta, \Phi} \big) = \mathsf{D}_{\mathsf{KL}}\big( P_{\Theta, \Phi, X_\Theta, Y_\Phi} \big\| Q_{\Theta, \Phi, X_\Theta, Y_\Phi} \big),$$

where the last step using the KL divergence chain rule. The proof is concluded by invoking the Donsker-Varadhan representation for KL divergence [33]

$$\mathsf{D}_{\mathsf{KL}}(P \| Q) = \sup_g \mathbb{E}_P[g] - \log \big( \mathbb{E}_Q[e^g] \big).$$

**Remark 9** (Max-sliced MI). *A similar variational form can be established for max-sliced MI, i.e.,* $\sup_{\theta, \phi} \mathsf{I}(\theta^\intercal X; \phi^\intercal Y)$. *In that case the variation representation is*

$$\sup_{g \in \mathcal{G}_{\mathsf{proj}}} \mathbb{E}\big[ g(X, Y) \big] - \log \Big( \mathbb{E}\big[ e^{g(\tilde{X}, \tilde{Y})} \big] \Big),$$

*with* $\mathcal{G}_{\mathsf{proj}} := \big\{ g \circ (\pi^\theta, \pi^\phi) : (\theta, \phi) \in \mathbb{S}^{d_x - 1} \times \mathbb{S}^{d_y - 1}, g : \mathbb{R}^2 \to \mathbb{R} \big\}$ *is the class of projecting functions. The derivation is similar and is thus omitted.*

## A.4   Proof of Theorem 1

Denote $\mathsf{I}_{X,Y}(\theta, \phi) := \mathsf{I}(\theta^\intercal X; \phi^\intercal Y)$ and notice that $\mathbb{E}\big[ \mathsf{I}_{XY}(\Theta, \Phi) \big] = \mathsf{SI}(X; Y)$, where $(\Theta, \Phi) \sim \mathsf{Unif}(\mathbb{S}^{d_x - 1}) \otimes \mathsf{Unif}(\mathbb{S}^{d_y - 1})$. By the triangle inequality we have

$$\big| \mathsf{SI}(X; Y) - \widehat{\mathsf{SI}}_{n,m} \big| \leq \bigg| \mathsf{SI}(X; Y) - \frac{1}{m} \sum_{i=1}^m \mathsf{I}_{XY}(\Theta_i, \Phi_i) \bigg| + \bigg| \frac{1}{m} \sum_{i=1}^m \mathsf{I}_{XY}(\Theta_i, \Phi_i) - \widehat{\mathsf{SI}}_{n,m} \bigg|.$$

For the first term, since $\{(\Theta_i, \Phi_i)\}_{i=1}^m$ are i.i.d., we obtain

$$\mathbb{E}\left[ \bigg| \mathsf{SI}(X; Y) - \frac{1}{m} \sum_{i=1}^m \mathsf{I}_{XY}(\Theta_i, \Phi_i) \bigg| \right] \leq \sqrt{\frac{1}{m} \mathsf{var}\big( \mathsf{I}_{XY}(\Theta, \Phi) \big)} \leq \frac{M}{2\sqrt{m}}$$

uniformly over $P_{X,Y} \in \mathcal{F}_d(M)$, where the last step follows because $0 \leq \mathsf{I}_{XY}(\Theta, \Phi) \leq \mathsf{I}(X;Y) \leq M$ a.s.

For the second term, recall the notation $X_\theta := \theta^\mathsf{T} X$ and $Y_\phi := \phi^\mathsf{T} Y$, and observe that

$$\mathbb{E}\left[\left|\frac{1}{m}\sum_{i=1}^{m}\mathsf{I}_{XY}(\Theta_i, \Phi_i) - \widehat{\mathsf{SI}}_{n,m}\right|\right] \leq \frac{1}{m}\sum_{i=1}^{m}\mathbb{E}\left[\left|\mathsf{I}_{XY}(\Theta_i, \Phi_i) - \hat{\mathsf{I}}_{XY}(\Theta_i, \Phi_i)\right|\right]$$
$$\leq \max_{\theta,\phi}\mathbb{E}\left[\left|\mathsf{I}(X_\theta; Y_\phi) - \hat{\mathsf{I}}\left(X_\theta^n, Y_\phi^n\right)\right|\right],$$

where $(X_\theta^n, Y_\phi^n)$ are pairwise i.i.d. samples of $(X_\theta, Y_\phi) \sim (\pi^\theta, \pi^\phi)_\sharp P_{X,Y}$. This falls under the MI risk bound from (5), yielding a bound of $\delta(n)$. $\qquad\square$

## A.5 Proof of Corollary 1

The bounded MI assumption in the definition of $\mathcal{F}_d(M)$ can be relaxed to a bounded the max-SMI, i.e.,

$$\max_{\theta\in\mathbb{S}^{d_x-1}, \phi\in\mathbb{S}^{d_y-1}}\mathsf{I}(\theta^\mathsf{T} X; \phi^\mathsf{T} Y) \leq M.$$

We next derive a uniform bound (over $(\theta, \phi) \in \mathbb{S}^{d_x-1} \times \mathbb{S}^{d_y-1}$) on

$$\mathsf{I}(\theta^\mathsf{T} X; \phi^\mathsf{T} Y) = h(\theta^\mathsf{T} X) + h(\phi^\mathsf{T} Y) - h(\theta^\mathsf{T} X, \phi^\mathsf{T} Y).$$

Since the Gaussian distribution maximize sliced (differential) entropy under a second moment constraint, we have

$$h(\theta^\mathsf{T} X) + h(\phi^\mathsf{T} Y) \leq \frac{1}{2}\log\left((2\pi e)^2(\theta^\mathsf{T}\Sigma_X\theta)(\phi^\mathsf{T}\Sigma_Y\phi)\right).$$

For the joint entropy, recall that log-concavity is preserved under affine transformations of coordinates and marginalization [34, Lemma 2.1]. Therefore $(\pi^\theta, \pi^\phi)_\sharp P_{X,Y}$ is also log-concave, and by Theorem 4 of [35] we obtain

$$h(\theta^\mathsf{T} X, \phi^\mathsf{T} Y) \geq \frac{1}{2}\log\left(\frac{e^4}{32}\left((\theta^\mathsf{T}\Sigma_X\theta)(\phi^\mathsf{T}\Sigma_Y\phi) - (\theta^\mathsf{T}\Sigma_{XY}\phi)(\phi^\mathsf{T}\Sigma_{YX}\theta)\right)\right).$$

Combining the two bounds we obtain

$$\mathsf{I}(\theta^\mathsf{T} X; \phi^\mathsf{T} Y) \leq \frac{1}{2}\log\left(\frac{\pi^2}{8}\frac{(\theta^\mathsf{T}\Sigma_X\theta)(\phi^\mathsf{T}\Sigma_Y\phi)}{(\theta^\mathsf{T}\Sigma_X\theta)(\phi^\mathsf{T}\Sigma_Y\phi) - (\theta^\mathsf{T}\Sigma_{XY}\phi)^2}\right)$$
$$= \frac{1}{2}\log\left(\frac{\pi^2}{8}\frac{1}{1 - \rho^2(\theta^\mathsf{T} X, \phi^\mathsf{T} Y)}\right)$$
$$\leq \frac{1}{2}\log\left(\frac{\pi^2}{8}\frac{1}{1 - \rho_{\mathsf{CCA}}^2(X,Y)}\right),$$

from which the claim follows. $\qquad\square$

## A.6 Proof of Corollary 2

The main idea is to use Theorem 2 from [26] to control the estimation error of each differential entropy in the decomposition of $\mathsf{I}(\theta^\mathsf{T} X; \phi^\mathsf{T} Y)$, where $(\theta, \phi) \in \mathbb{S}^{d_x-1} \times \mathbb{S}^{d_y-1}$. To that end, we first show that since $p_{X,Y} \in \mathsf{Lip}_{s,p,d_x+d_y}(L)$ (by assumption), any of its projections also belong to a generalized Lipschitz class as well of the appropriate dimension. To state the result, let $p_{X_\theta}, p_{Y_\phi}$ and $p_{X_\theta,Y_\phi}$ be the density of $\theta^\mathsf{T} X$, $\phi^\mathsf{T} Y$, and $(\theta^\mathsf{T} X, \phi^\mathsf{T} Y)$, respectively.

**Lemma 1** (Lipschitzness of projections). *If $p_{X,Y} \in \mathsf{Lip}_{s,p,d_x+d_y}(L)$, then $p_{X_\theta}, p_{Y_\phi} \in \mathsf{Lip}_{s,p,1}(L)$, and $p_{X_\theta,Y_\phi} \in \mathsf{Lip}_{s,p,2}(L)$, for any $(\theta, \phi) \in \mathbb{S}^{d_x-1} \times \mathbb{S}^{d_y-1}$.*

*Proof.* We present the derivation for $p_{X_\theta,Y_\phi}$; the proof for $p_{X_\theta}$ and $p_{Y_\phi}$ is similar. Note that Definition 4 is invariant to rotations of both the $X$ and $Y$. Hence, without loss of generality,

we may assume that $\theta$ and $\phi$ are both canonical unit vectors, e.g., both equal $e_1 = (1\ 0\ \ldots 0)$ of the appropriate dimension. Consequently, $\theta^\mathsf{T} X = X_1$ and $\phi^\mathsf{T} Y = Y_1$. Denote $x_{2:} := (x_2 \ldots x_{d_x})$ and $y_{2:} := (y_2 \ldots y_{d_y})$ and write

$$p_{X_\theta, Y_\phi}(x_1, y_1) = \int_{[0,1]^{d'}} p_{X,Y}(x, y)\mathrm{d}x_{2:}\mathrm{d}y_{2:},$$

where $d' = d_x + d_y - 2$ and we have used the fact that $\theta^\mathsf{T} X = X_1$ and $\phi^\mathsf{T} Y = Y_1$. Finally, for each $x_1, y_1 \in [0,1]^2$, we denote $p^{(x_1, y_1)}(x_{:2}, y_{:2}) := p_{X,Y}(x_1, x_{:2}, y_1, y_{:2})$.

We now bound the norms that make up the definition of the generalized Lipschitz class. First, consider

$$\|p_{\theta,\phi}\|_{p,2} = \left\| \int_{[0,1]^{d'}} p^{(\cdot,\cdot)}(x_{:2}, y_{:2})\mathrm{d}x_{:2}\mathrm{d}y_{:2} \right\|_{p,2}$$

$$\leq \left( \int_{[0,1]^2} \left( \int_{[0,1]^{d'}} \left( p^{(x_1, y_1)}(x_{:2}, y_{:2}) \right)^p \mathrm{d}x_{:2}\mathrm{d}y_{:2} \right) \mathrm{d}x_1 \mathrm{d}y_1 \right)^{1/p}$$

$$= \|p_{X,Y}\|_{p, d_x + d_y},$$

where the 2nd step follows because $\int_{[0,1]^{d'}} p^{(x_1, y_1)}(x_{:2}, y_{:2})\mathrm{d}x_{:2}\mathrm{d}y_{:2} \leq \left\| p^{(x_1, y_1)} \right\|_{p, d'}$ by Jensen's inequality. Similarly, denoting by $e \in \mathbb{R}^d$ the vector that has 1's in its first and $(d_x + 1)$th coordinates and 0's otherwise, for any $(x_1, y_1) \in [0,1]^2$, we have

$$\left| \Delta^r_{t(1\ 1)} p_{\theta,\phi}(x_1, y_1) \right| \leq \int_{[0,1]^{d'}} \left| \Delta^r_{te} p^{(x_1, y_1)}(x_{:2}, y_{:2}) \right| \mathrm{d}x_{:2}\mathrm{d}y_{:2} \leq \left\| \Delta^r_{te} p^{(x_1, x_2)} \right\|_{p, d'},$$

where the last step uses Jensen's inequality once more. Having that, we obtain

$$\|\Delta^r_{te} p_{\theta,\phi}\|_{p,2} \leq \left( \int_{[0,1]^2} \left\| \Delta^r_{te} p^{(x_1, y_1)} \right\|^p_{p, d'} \mathrm{d}x_1 \mathrm{d}y_1 \right)^{1/p} = \|\Delta^r_{te} p_{X,Y}\|_{p, d_x + d_y}.$$

Consequently $\|p_{\theta,\phi}\|_{\mathsf{Lip}_{p,s,2}} \leq \|p_{X,Y}\|_{\mathsf{Lip}_{p,s,d_x+d_y}} \leq L$, for all $(\theta, \phi) \in \mathbb{S}^{d_x-1} \times \mathbb{S}^{d_y-1}$, as required. $\qquad \square$

Based on the lemma, we may invoke [26, Theorem 2] to obtain error bounds on the estimation of the sliced entropy terms that comprise SMI. We first restate the result of [26]: if $X \sim p_X \in \mathsf{Lip}_{p,s,d}(L)$, for $d \in \mathbb{N}$, $s \in (0, 2]$, $p \in [2, \infty)$, is $\beta$-sub-Gaussian[5], $\beta > 0$, and satisfies the tail bound $\int_{\mathbb{R}^d} e^{\beta \|x\|^2} p_X(x)\mathrm{d}x \leq L$, then

$$\left( \mathbb{E}\left[ \left( \hat{\mathsf{H}}(X^n) - \mathsf{H}(X) \right)^2 \right] \right)^{\frac{1}{2}} \leq C \left( (n \log n)^{-\frac{s}{s+d}} (\log n)^{\frac{d}{2}\left(1 - \frac{d}{p(s+d)}\right)} + n^{-\frac{1}{2}} \right), \qquad (17)$$

for a constant $C$ depending only on $s, p, d, \beta, L$.

Note that $p_{X_\theta}$, $p_{X_\theta, Y_\phi}$, and $p_{Y_\phi}$, for any $(\theta, \phi) \in \mathbb{S}^{d_x-1} \times \mathbb{S}^{d_y-1}$, are compactly supported and hence sub-Gaussian (with a sub-Gaussian constant and tail bound that depend only on $d$ and $L$). Lemma 1 then implies that $\mathsf{H}(\theta^\mathsf{T} X)$, $\mathsf{H}(\phi^\mathsf{T} Y)$, and $\mathsf{H}(\theta^\mathsf{T} X, \phi^\mathsf{T} Y)$ can all be estimated within the framework of [26] under the error bound from (17). Denoting the respective estimators by adding a hat to the differential entropy notation and letting $\mathsf{e}_\theta$, $\mathsf{e}_\phi$, and $\mathsf{e}_{\theta,\phi}$ be their $L_2$ errors, we obtain

$$\max \left\{ \mathsf{e}_\theta, \mathsf{e}_\phi, \mathsf{e}_{\theta,\phi} \right\} \leq C \left( (n \log n)^{-\frac{s}{s+2}} (\log n)^{\left(1 - \frac{2}{p(s+2)}\right)} + n^{-\frac{1}{2}} \right), \quad \forall (\theta, \phi) \in \mathbb{S}^{d_x-1} \times \mathbb{S}^{d_y-1}. \tag{18}$$

Here we used the fact that the rate is dominated by the error in estimating the 2-dimensional differential entropy $\mathsf{H}(\theta^\mathsf{T} X, \phi^\mathsf{T} Y)$. Recall that the considered MI estimator relies on the decomposing

$$\mathsf{I}(\theta^\mathsf{T} X' \phi^\mathsf{T} Y) = \mathsf{H}(\theta^\mathsf{T} X) + \mathsf{H}(\phi^\mathsf{T} Y) - \mathsf{H}(\theta^\mathsf{T} X, \phi^\mathsf{T} Y)$$

and estimating each sliced entropy separately. Bounding the MI estimation error via (18) produces the result. $\qquad \square$

---

[5] A $d$-dimensional random variable $X$ is $\beta$-sub-Gaussian if $\mathbb{E}\left[ e^{\beta \|X\|^2} \right] < \infty$.

## A.7 Proof of Proposition 4

**Proof of 1.** By Part 2 of Proposition 1, we have

$$
\mathsf{SI}(\mathrm{A}_x X + b_x; \mathrm{A}_y Y + b_y) \leq \sup_{\theta,\phi} \mathsf{I}\big(\theta^{\mathsf{T}}(\mathrm{A}_x X + b_x); \phi^{\mathsf{T}}(\mathrm{A}_y Y + b_y)\big)
$$
$$
\leq \sup_{\theta,\phi} \mathsf{I}(\theta^{\mathsf{T}} X; \phi^{\mathsf{T}} Y),
$$

where in the last step we have used the DPI of classic MI. Now, let $\{(\theta_i, \phi_i)\}_{i=1}^{\infty}$ be a sequence converging to the supremum of $\mathsf{I}(\theta^{\mathsf{T}} X; \phi^{\mathsf{T}} Y)$. Set $b_y = b_x = 0$, and consider the sequence $\{(\mathrm{A}_x^i, \mathrm{A}_y^i)\}_{i=1}^{n}$ where $\mathrm{A}_x^i = (\theta_i \ 0 \ \dots \ 0)^{\mathsf{T}}, \mathrm{A}_y^i = (\phi_i \ 0 \ \dots \ 0)^{\mathsf{T}}$. Clearly, for each $i$, we have

$$
\mathsf{SI}(\mathrm{A}_x^i X; \mathrm{A}_y^i Y) = \mathsf{I}(\theta_i^{\mathsf{T}} X; \phi_i^{\mathsf{T}} Y),
$$

which implies the first claim.

**Proof of 2.** Let $\mathcal{O}(d)$ be the set of orthogonal $d \times d$ real-valued matrices. For $\mathrm{U} \sim \mathsf{Unif}\big(\mathcal{O}(d)\big)$ and $\tilde{\Theta} \sim \mathsf{Unif}(\mathbb{S}^{r-1})$ independent, note that $[\mathrm{U}]_{:,1:r}\tilde{\Theta} \sim \mathsf{Unif}(\mathbb{S}^{d-1})$, where $[\mathrm{U}]_{:,1:r}$ stands for the first $r$ columns of $\mathrm{U}$. We therefore have:

$$
\mathsf{SI}(\mathrm{A}_x X; \mathrm{A}_y Y) = \mathsf{I}\big(\tilde{\Theta}^{\mathsf{T}}[\mathrm{U}_x]_{:,1:r_x}^{\mathsf{T}} \mathrm{A}_x X; \tilde{\Phi}^{\mathsf{T}}[\mathrm{U}_y]_{:,1:r_y}^{\mathsf{T}} \mathrm{A}_y Y \big| \tilde{\Theta}, \tilde{\Phi}, \mathrm{U}_x, \mathrm{U}_y\big)
$$
$$
\leq \sup_{\substack{\mathrm{U}_x \in \mathcal{O}(d_x), \\ \mathrm{U}_y \in \mathcal{O}(d_y)}} \mathsf{SI}([\mathrm{U}_x]_{:,1:r_x}^{\mathsf{T}} \mathrm{A}_x X; [\mathrm{U}_y]_{:,1:r_y}^{\mathsf{T}} \mathrm{A}_y Y), \tag{19}
$$

where the last inequality follows by upper bounding the expectation by the supremum and the independence of $(\mathrm{U}_x, \mathrm{U}_y)$ and $(\tilde{\Theta}, \tilde{\Phi}, X, Y)$.

Note that if $\mathrm{A}_x \in \mathcal{M}_{d_x, d_x}(r_x, c_x)$ and $\mathrm{A}_y \in \mathcal{M}_{d_y, d_y}(r_y, c_y)$, then $[\mathrm{U}_x]_{:,1:r_x}^{\mathsf{T}} \mathrm{A}_x \in \mathcal{M}_{r_x, d_x}(r_x, c_x)$, $[\mathrm{U}_y]_{:,1:r_y}^{\mathsf{T}} \mathrm{A}_y \in \mathcal{M}_{r_y, d_y}(r_y, c_y)$ (since the first $r$ singular values of $\mathrm{A}_x$ and $\mathrm{A}_y$ are inside $[1/c_x, c_x]$ and $[1/c_y, c_y]$, respectively). Using this observation while supremizing the LHS of (19), we obtain

$$
\sup_{\substack{\mathrm{A}_x \in \mathcal{M}_{d_x, d_x}(r_x, c_x), \\ \mathrm{A}_y \in \mathcal{M}_{d_x, d_x}(r_y, c_y)}} \mathsf{SI}(\mathrm{A}_x X; \mathrm{A}_y Y) \leq \sup_{\substack{\mathrm{B}_x \in \mathcal{M}_{r_x, d_x}(r_x, c_x), \\ \mathrm{B}_y \in \mathcal{M}_{r_y, d_y}(r_y, c_y)}} \mathsf{SI}(\mathrm{B}_x X; \mathrm{B}_y Y).
$$

The opposite inequality follows by only considering those matrices $(\mathrm{A}_x, \mathrm{A}_y)$ whose bottom $d_x - r_x$ or $d_y - r_y$ rows are zeros.

## A.8 Proof of Corollary 3

We begin by considering fixed $\mathrm{W}_x, \mathrm{W}_y, b_x, b_y$. By Part 2 of Proposition 1, we have

$$
\mathsf{SI}(\mathrm{A}_x \sigma(\mathrm{W}_x^{\mathsf{T}} X + b_x); \mathrm{A}_y \sigma(\mathrm{W}_y^{\mathsf{T}} Y + b_y)) \leq \sup_{\theta,\phi} \mathsf{I}\big(\theta^{\mathsf{T}} \mathrm{A}_x \sigma(\mathrm{W}_x^{\mathsf{T}} X + b_x); \phi^{\mathsf{T}} \mathrm{A}_y \sigma(\mathrm{W}_y^{\mathsf{T}} Y + b_y)\big)
$$
$$
\leq \sup_{\theta,\phi} \mathsf{I}\big(\theta^{\mathsf{T}} \sigma(\mathrm{W}_x^{\mathsf{T}} X + b_x); \phi^{\mathsf{T}} \sigma(\mathrm{W}_y^{\mathsf{T}} Y + b_y)\big), \tag{20}
$$

where in the last step we have used the DPI of classic MI. Now, let $\{(\theta_i, \phi_i)\}_{i=1}^{\infty}$ be a sequence converging to the supremum of $\mathsf{I}\big(\theta^{\mathsf{T}} \sigma(\mathrm{W}_x^{\mathsf{T}} X + b_x); \phi^{\mathsf{T}} \sigma(\mathrm{W}_y^{\mathsf{T}} Y + b_y)\big)$. Consider the sequence $\{(\mathrm{A}_x^i, \mathrm{A}_y^i)\}_{i=1}^{n}$ where $\mathrm{A}_x^i = (\theta_i \ 0 \ \dots \ 0)^{\mathsf{T}}, \mathrm{A}_y^i = (\phi_i \ 0 \ \dots \ 0)^{\mathsf{T}}$. Clearly, for each $i$, we have

$$
\mathsf{SI}\big(\mathrm{A}_x^i \sigma(\mathrm{W}_x^{\mathsf{T}} X + b_x); \mathrm{A}_y^i \sigma(\mathrm{W}_y^{\mathsf{T}} Y + b_y)\big) = \mathsf{I}\big(\theta_i^{\mathsf{T}} \sigma(\mathrm{W}_x^{\mathsf{T}} X + b_x); \phi_i^{\mathsf{T}} \sigma(\mathrm{W}_y^{\mathsf{T}} Y + b_y)\big),
$$

which implies that equality in (20) can be achieved. Hence the supremum of the LHS over $A_x, A_y$ equals the RHS. Supremizing both sides over $\mathrm{W}_x, \mathrm{W}_y, b_x, b_y$ then yields the corollary.

## B Pseudocode and Complexity of the SMI Estimator

Algorithm 1 shows the pseudocode for our SMI estimator (6), repeated here:

$$
\widehat{\mathsf{SI}}_{n,m} = \widehat{\mathsf{SI}}_{n,m}(X^n, Y^n, \Theta^m, \Phi^m) := \frac{1}{m} \sum_{i=1}^{m} \hat{\mathsf{I}}\big((\Theta_i^{\mathsf{T}} X)^n, (\Phi_i^{\mathsf{T}} Y)^n\big).
$$

---

**Algorithm 1** SMI Estimator

---

**Require:** $n$ (pairs of) samples $(X^n, Y^n)$ i.i.d. according to $P_{X,Y} \in \mathcal{P}(\mathbb{R}^{d_x} \times Y \in \mathbb{R}^{d_y})$, a scalar
    MI estimator $\hat{\mathsf{I}}(\cdot; \cdot)$, and a chosen number of slices $m$.
    **for** $i = 1 : m$ **do**
        Sample $\Theta_i$ uniform on the sphere $\mathbb{S}^{d_x - 1}$.[6]
        Sample $\Phi_i$ uniform on the sphere $\mathbb{S}^{d_y - 1}$.
        Compute the MI estimate: $S_i \leftarrow \hat{\mathsf{I}}\big((\Theta_i^\mathsf{T} X)^n, (\Phi_i^\mathsf{T} Y)^n\big)$.
    **end for**
    $\widehat{\mathsf{SI}}_{n,m} \leftarrow \frac{1}{m} \sum_{i=1}^{m} S_i$

---

It requires as input some 1 dimensional MI estimator $\hat{\mathsf{I}}(\cdot; \cdot)$ which takes as input a sample from the joint distribution of two 1-dimensional variables and outputs an estimate of their MI.

Reading off from Algorithm 1, the computational complexity of the estimator is $O\big(m(d_x + d_y)n + mA(n)\big)$, where $A(n)$ is the computational complexity of the scalar MI estimator. It can be seen that the computational complexity scales linearly with dimension and the number of slices $m$. The scaling with the number of samples $n$ follows $\max\{n, A(n)\}$.

## C MI Convergence Experiment

In Figure 5, we show convergence results of MI estimation using the Kozachenko-Leonenko, EDGE [16], and MINE [29] estimators. The data is the standard Gaussian vectors with 5 overlapping components as described for the $d = 10$ case in Figure 1(b,c) of the main text. Note that the MI estimators converge slowly in this high dimensional regime, in contrast to the $n^{-1/2}$ convergence rate for SMI estimation seen in Figure 1(b).

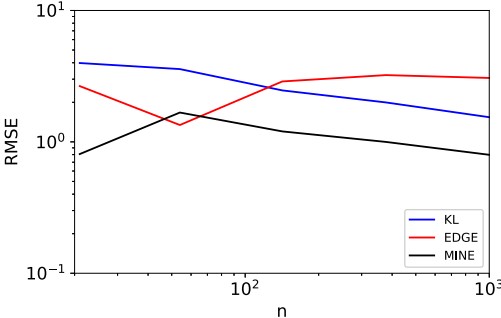

Figure 5: Convergence of MI estimation (via Kozachenko-Leonenko, EDGE, and MINE estimators) versus the number of data samples $n$ for $d = 10$ standard Gaussian vectors with 5 overlapping entries. Note that the convergence is significantly slower than that in the SMI estimation experiment from Figure 1(b).

---

[6]A uniform sample from $\mathbb{S}^{d_x - 1}$ can be found by sampling a vector $Z$ from a $d_x$-dimensional isotropic Gaussian and forming $Z / \|Z\|_2$.