# OpenReview forum: "Sliced Mutual Information: A Scalable Measure of Statistical Dependence"
_NeurIPS.cc/2021/Conference — NeurIPS 2021 Spotlight_

### Official Review · Reviewer_dcfo · 2021-07-11

**Rating:** 6
**Confidence:** 4

**Summary:**

The paper proposes a measure of statistical dependence called SMI (sliced MI) defined as the MI between 1-dimensional projections using random unit vectors. The paper shows that SMI preserves many properties of MI like nonnegativity (zero iff independent) but not all (notably DPI, which the paper argues as a good thing). The upshot is that estimation of SMI can be done with any 1-dimensional MI estimator and an MC integrator, and that this can have better sample efficiency than direct MI estimation when the observations are high-dimensional but have a low-dimensional structure.

**Limitations And Societal Impact:**

Contextualizing SMI in well-known limitations of estimating mutual information would be useful.

**Main Review:**

STRENGTHS

- SMI is a simple and intuitive quantity. It seems like a natural addition to existing slicing techniques of measuring the Wasserstein distance.

- The paper is quite thorough in establishing properties of SMI, including not only nonnegativity and a KL formulation but also the chain rule and a DV-like variational form.

- The paper also provides convergence guarantees that relate the error of the SMI estimator to the error of a 1-dimensional MI estimator. The utility of SMI in extracting features based on the violation of DPI is also discussed.


WEAKNESSES

- SMI is a lower bound on MI and cannot be larger than the log sample size [1], so it doesn't seem to offer much advantage when the underlying MI is large which is really the setting that we care about. SMI seems mostly useful in a high-dimensional setting with a low-dimensional structure (with small MI). There are already many available low-variance high-bias estimators of MI when MI is small [2], so I'm a bit unsure how significant or impactful SMI will be in practice other than being a theoretically interesting quantity.

- I feel experiments should consider more than 1 MI estimator. There are many neural MI estimators with different behaviors [2], [1] which might change the story in Figure 2. As for feature extraction, experiments should ideally compare with the existing literature like MINE, DIM, BYOL, etc. in downstream performance, but I understand that it's probably out of scope so this is more minor.


[1] Formal Limitations on the Measurement of Mutual Information

[2] On Variational Bounds of Mutual Information


**Time Spent Reviewing:**

2.5

---

> ### Author Response · Authors · 2021-08-09
> **Response to Reviewer dcfo**
>
> We thank the reviewer for the constructive feedback. Below we address the specific comments/concerns raised.
>
> 1. __SMI works well given low-dimensional structure and large MI versus SMI values:__ The reviewers mentions at the end of the summary that SMI 'can have better sample efficiency than direct MI estimation when the observations are high-dimensional but have a low-dimensional structure.' It is then further suggested that SMI is useful only when MI itself is small, in which case good MI estimators are already available.
>
>     First, we would like to mention that there is no intrinsic low-dimensionality assumption in the paper. The empirical convergence rates from Section 3.1 are all derived without imposing any low-dimensional structure on the population distribution. It may be the case that Figure 2 gave the reviewer the impression that the gain of using SMI instead of MI is highest in such scenarios (namely, the comparable performance in subfigure (d)). While this is true in that particular example, we do not expect gains to be restricted to such setups in general. For instance, the independence assumption makes MI easy to evaluate in Setting (d) via tensorization, but intrinsically high-dimensional variables need not have independent coordinates. In such cases, SMI still offers a scalable measure of dependence that can be computed with relative ease, compared to the costly estimation of MI.
>
>     Furthermore (copying from the 1st response to __R-C7hp__ for the reviewer's convenience): We would like to kindly stress that SMI is *not* proposed as an estimate of MI but rather as a new measure of dependence between random variables. The theoretical exploration of SMI properties (Section 3) aims to justify it as a meaningful such measure, while the statistical study (Section 3.1) shows that SMI can be estimated efficiently even in high-dimensional settings (where MI estimation may be infeasible). In particular, SMI cannot be larger than the value of the largest MI term across 1D slices, which may have little to do with dimension in general. We will add a discussion on the above points in the revision.
>
> 2. __Other MI estimators:__ While we chose the KL estimator because of its popularity and theoretical results showing it is near minimax optimal, in the revision we will provide results for additional SMI and MI estimators. Copying the 3nd and 4th responses to __R-C7hP__ for convenience: Specifically, we will add estimation error convergence rates for high-dimension MI estimation via the kNN method from [15], the EDGE estimator from [16], as well as MINE [17]. In addition, convergence rates for SMI estimation via S-MINE (see Section 4.3) will be added to the current kNN-based estimate plots for completeness. Rates for these methods will be shown for several dimensions with the goal of empirically validating the scalability to high-dimensions versus lack thereof of SMI versus MI, respectively. We will rerun the above experiment for MNIST data and expect to observe similar trends.
>
>     If the reviewer would like us to also include SMI estimates based on a scalar MI estimator other than KL, please let us know which one and we are happy to do so.
>
> 3. __Other feature extractors:__ We agree with the reviewer that adding downstream DIM, BYOL, etc. experiments are beyond the scope of this 9 page paper. Our paper is primarily a theoretical work aiming to inspire practical followup works of exactly that type (indeed we are intensely interested in exploring such applications ourselves). Proof-of-concept experiments are included here for illustration of the theoretical bounds; not (yet) as proof of practical engineering usefulness.

---

### Official Review · Reviewer_9G8T · 2021-07-12

**Rating:** 7
**Confidence:** 4

**Summary:**

The authors introduce the Sliced Mutual Information (SMI) which is a measure of dependence between (sets) of random variables X and Y. The measure is defined as the average mutual information I(\thetaX; \phiY) among all 1-dimensional projections (slices) \thetaX and \phiY from the unit sphere. In practice SMI is computed with m slices sampled uniformly from the unit sphere combined with any empirical estimator, e.g., the Kozachenko-Leonenko. SMI overcomes the statistical inefficiency of mutual information in high-dimensional spaces while maintaining many of its properties, e.g., non-negativity and chain rule.  However, SMI violates the Data Processing Inequality (DPI). In addition, the authors prove the consistency of SMI and argue that it can be used as an objective to extract informative features by fitting transformation matrices. The results show that SMI is indeed consistent, can be used to identify statistical dependence, and can be used to extract informative features.

**Limitations And Societal Impact:**

The authors do not discuss the limitations of their method, nor in what scenarios it is preferable over other techniques, e.g., MINE.


**Main Review:**

The idea to build a dependence measure as the average of all 1-dimensional projections combined with mutual information is novel (to the best of my knowledge). It is also true that mutual information can be problematic in the high-dimensional settings. SMI is therefore a good alternative of mutual information both in terms of computational and statistical efficiency. In addition SMI can be used as an objective to find data transformations that increase SMI (as opposed to the standard definition). This way feature extraction is obtained.

While I think that SMI is overall a good contribution, I do have some comments:
- it is unclear when the original formulation (i.e., averaging over all 1-dim projections) should be preferred over the extraction-based one. Or in other words, which one to use and when? To me the second formulation seems to be superior because the first one can arrive at a misleading result (as is the case in the MNIST experiment). Perhaps the authors intent to use the base formulation as a stepping stone to the extraction one, but then this should be stated more clear in the text.
- isn't the maximum definition superior to the average? When should one used and when should the other? From Prop. 4 we have that the supremum of mutual information with 1-dim projections is equal to the supremum of SMI over all transformation matrices. This combined with the MNIST experiment (where the average definition leads to a small SMI score) make it look like the maximum should be preferred.
- in general it is unclear in what scenarios the difference versions of SMI are best applicable
- since the authors refer to MINE, why isn't it used in the comparison experiments to get a better understanding of SMI capabilities (e.g., 4.1)? If statistical inefficiency in high-dimensions is the problem, doesn't MINE solve that already?

Some notes:
- perhaps it will be easier for the audience that is not trained in projections if the authors explain somewhat better the matrix notation they use. For example, the design matrix X seems to be transposed by default (compared to the general scenario where X is NxD, with N the number of samples and D the dimensions). Also, are vectors by default column vectors? Also, in 4.2.a is 1^TX a vector added to matrix Z?
- in the example in 3.2, Y should most likely be equal to X1 and not to X2 (or in general there seems to be something wrong in this paragraph)
- the independence testing evaluation looks more like statistical power experiment since the simulated Y are a function of the input variables. An independence testing experiment would imply that SMI is used to detect independence when independence is true. Perhaps the authors can change the title to Detecting dependence (or something along these lines).
- scalability is misleading: does it refer to computational scalability or to statistical? For example, Figure 2 shows statistical scalability. This can be rephrased to statistical efficiency for clarity



**Time Spent Reviewing:**

15

---

> ### Author Response · Authors · 2021-08-09
> **Response to Reviewer 9G8T**
>
> We thank the reviewer for the constructive and positive feedback. Below we address the specific comments/concerns raised.
>
> 1. __Average versus 'extraction-based' SMI:__ The reviewer asked when the average SMI formulation (Definition 1) should be preferred over the 'extraction-based one'. We are not completely clear on what the latter refers to, and would be glad to get a clarification. If the reviewer refers to the LHS of Equation (11), then the question is essentially about $\mathsf{SI}(X;Y)$ versus $\mathsf{mSI}(X;Y):=\sup_{\theta,\phi}\mathsf{I}(\theta^\intercal X;\phi^\intercal Y)$, where the latter can be termed max SMI (mSMI) for the purpose of this response. We first mention that the current submission only proposes and studies $\mathsf{SI}(X;Y)$ as a measure of dependence, with mSMI used only as an auxiliary quantity for that study.
>
>     That said, mSMI is a valid quantity to consider---we plan to explore it in future work and expect it to share many of the properties of SMI. However, two distinctions are in order: (i) the average version is cheaper to compute (via MC averaging), as it involves no optimization; (ii) we anticipate that, due to the optimization over a $d$-dimensional sphere, estimating mSMI would require a number of samples that grows with dimension; and (iii) while $\mathsf{SI}(X;Y)$ can be used for feature extraction, Proposition 4 effectively shows that mSMI is not useful for that purpose since $\sup_{\mathrm{A}_x,\mathrm{A}_y,b_x,b_y} \mathsf{mSI}(\mathrm{A}_x X + b_x; \mathrm{A}_y Y + b_y)=\mathsf{mSI}(X;Y)$ and invariance, e.g., to linear extractors, is recovered.
>
>     Regarding the MNIST experiment, we point out that mSMI will *always* be at least as large as SMI (by definition), but this does not imply that the mSMI is 'superior' since SMI is not meant as an estimator of MI. Instead, the point of the MNIST experiment is to show that optimizing over a feature extractor to maximize SMI succeeds in finding a small number of features that have high MI.
>
>     Overall, both SMI and mSMI are valid and interesting measures of dependence. They are expected to share many properties but also differ in some aspects. As such, it is hard to pose one as globally superior/inferior, and the preference should be guided by the intended application. We will address the above in the summary section of the revised manuscript and note that the current paper only proposes/focuses on a single SMI formulation. We hope this addresses the 1st and 2nd comments of the reviewer, but please let us know if further clarification is needed.
>
>
>
> 2. __Comparison to MINE:__ Thank you for that suggestion. We agree that it is interesting to show that MINE, along with other MI estimators, suffers from poor scalability to high dimensions. Copying the 3nd and 4th responses to __R-C7hP__ for convenience: Specifically, we will add estimation error convergence rates for high-dimension MI estimation via the kNN method from [15], the EDGE estimator from [16], as well as MINE [17]. In addition, convergence rates for SMI estimation via S-MINE (see Section 4.3) will be added to the current kNN-based estimate plots for completeness. Rates for these methods will be shown for several dimensions with the goal of empirically validating the scalability to high-dimensions versus lack thereof of SMI versus MI, respectively. We will rerun the above experiment for MNIST data and expect to observe similar trends.
>
>     We would also like to mention that, generally, MINE does not have strong theoretical guarantees, and is known to underperform when MI is high (see [D. McAllester and K. Stratos, ``Formal limitations on the measurement of mutual information.'' International Conference on Artificial Intelligence and Statistics, 2020]). Thus, when $\mathsf{I}(X,Y)$ is large and MI estimation via MINE is ill-advised, SMI can still be used to obtain a meaningful measure of dependence between these random variables under strong accuracy guarantees. As MI often grows with dimensions, and bearing in mind the high dimensionality of real-world data, we expect SMI to be useful in a variety of applications. Also, as pointed out in our paper, $n^{-1/(d_x+d_y)}$ is a minimax rate for MI estimation under the assumptions stated. MINE (or any other MI estimator) therefore cannot exceed that rate, at least in the worst case. Lastly, since SMI is not an estimator of MI, MINE is not directly comparable to it.
>
>
> 3. __Clarify notation:__ We thank the reviewer for this comment. Vectors are indeed assumed to be column vectors by default, and $X$ is a random vector with values in $\mathbb{R}^{d\times 1}$, not a design matrix. In 4.2(a), $\mathbf{1}^\intercal X$ is thus a scalar. There was a typo in the definition of Y in case (a), which should have been $Y = \frac{1}{\sqrt{2}}\big (\frac{1}{d}(\mathbf{1}^\intercal X)\mathbf{1}+Z\big)$, where $Z$ is a $d$-dimensional Gaussian. This typo is fixed now.
>
> 4. __Typo in the example from Section 3.2:__ The reviewer is correct, there is indeed a typo in that example: $Y$ should be $X_1$ and $Z$ should be $X_2$. This will be fixed in the revision. Thank you for noticing this.
>
> 5. __Independence testing or statistical power:__ We thank the reviewer for this constructive comment and acknowledge that we did not make the experiment setup sufficiently clear. Rather than simply testing statistical power, in Figure 2 we generate $n$-sample datasets for each of 200 pairs of the dependent $(X,Y)$ pairs along with 200 copies of *independent* $(X,Y)$ pairs with the same marginal distributions. The SMI is then estimated for each of the 400 pairs, and the independence test thresholds the SMI, with independence declared when SMI is below the threshold. Rather than choosing a single threshold, we report the Area Under the ROC Curve, which is a standard way to illustrate performance of detectors. The ROC curve is found in the regular way, by varying over all possible thresholds and plotting the probabilities of *both types* of error (false alarm and missed detection) for each threshold. We will add these details to the text.
>
> 6. __Scalability is misleading:__ What we mean by 'scalability' is that SMI can be efficiently estimated in arbitrary dimension, which only concerns the statistical aspects. We will make this clear this in the revision and adopt the term 'statistical efficiency', as suggested.
>
>
> 7. __Discuss limitations and societal impact:__ Thank you for bringing up this important and valid point. As this was brought up uniformly by all reviewers, we kindly copy our response to __R-C7hp__ above: We will add a discussion about SMI being a new measure of dependence between random variables that (although related) is different from classic MI. We will also highlight potential negative impacts of treating SMI as an estimate of MI and provide the above example to support that. Please also see Item 1 from the response to __R-C7hp__ for an example concerning the gap between SMI and MI.
>
>     Addressing MINE in particular, we recall that it tends to underperform in cases where MI is high. SMI, on the other hand, can still be used to obtain a meaningful measure of dependence between random variables in that regime, under strong accuracy guarantees. High-dimensional settings are typical in practice, and so we expect SMI to be useful in a variety of applications.

---

### Official Review · Reviewer_C7hP · 2021-07-14

**Rating:** 7
**Confidence:** 3

**Summary:**

This paper proposes sliced mutual information (SMI), which is a new estimator of the mutual information (MI) particularly effective for high dimensional settings. The definition of SMI is simple and its computation is efficient, and it has desirable theoretical properties. Empirical evaluation shows the effectiveness of the SMI compared to the estimated MI.


**Limitations And Societal Impact:**

The authors do not provide the limitations and potential negative societal impact. I would suggest discussing the limitation of the proposal; that is, when it fails to estimate the MI, and discuss the negative impact of the proposal if such a situation occurs.

**Main Review:**

### Originality

The originality of this paper is high as it proposes a simple yet effective new MI estimator.
It would be better if the difference of the proposal and existing MI estimators are discussed more carefully.

### Quality

This paper is well written and the overall quality is high.
- In particular, this paper performs thorough theoretical analysis of the SMI and its estimator, which is impressive and a good contribution.
- The pseudo-code of the estimator of the SMI and its complexity analysis would be desirable for better readability (at least in Appendix), although it is relatively clear from its definition.
- In contrast, empirical evaluation is not thorough. I have the following comments:
    - Although several MI estimators are mentioned in Introduction ([14-18]), there is no comparison with such methods in experiments. The effectiveness of the proposed method is more convincing if such empirical comparison is performed.
    - In Figure 1, is the RMSE an error between the SMI and the MI? There is no clear definition in the paper.
    - In addition, in Figure 1, since generated data here is quite simple, I am interested in how the convergence rate is for other types of datasets?
    - In Section 4.2, only relatively simple relationships ((a)-(d)) are examined. Since the MI can measure any statistical dependence, comparison on various types of relationships is desirable (e.g. relationships used in Reshef et al., Detecting Novel Associations in Large Data Sets, Science, 2011).


### Clarity
This paper is clearly written and easy to follow.

### Significance
The proposed method is expected to be widely applicable in machine learning and data science as the proposal is simple. Hence the significance of the contribution is high.


**Time Spent Reviewing:**

3

---

> ### Author Response · Authors · 2021-08-09
> **Repy to Reviewer C7hP**
>
> We thank the reviewer for the constructive and positive feedback. Below we address the specific comments/concerns raised.
>
> 1. __SMI as an estimate of MI:__ The reviewer asked about comparison between SMI and existing MI estimators. We would like to kindly stress that SMI is *not* proposed as an estimate of MI but rather as a new measure of dependence between random variables. The theoretical exploration of SMI properties (Section 3) aims to justify it as a meaningful such measure, while the statistical study (Section 3.1) shows that SMI can be estimated efficiently even in high-dimensional settings. We will highlight this point in the revision to avoid any potential confusion, stressing that SMI should not be viewed as a proxy of MI. We will also discuss the hazard in adopting such a perspective and explain that the difference between SMI and MI can be arbitrarily large. To see this (copying from our response above for the reviewer's convenience), consider the example from the beginning of Section 3.2 (note that there was a typo in the submitted version; the correct setup has $Y=X_1$ and no $Z$), where for any $0<a<\infty$ we have $\mathsf{I}\big(g_a(X);Y\big)=\infty$ while $\mathsf{SI}\big(g_a(X);Y\big)$ is finite.
>
> 2. __Pseudo-code of the estimator and complexity analysis:__ We will add a pseudo-code of the estimator to the supplement along with a comment on its computational complexity ($O(m (d_x+d_y) n + m A(n))$ where $A(n)$ is the computational complexity of the MI estimator used to compute the the MI between 1D slices).
>
> 3. __Comparison between SMI and classic MI empirical convergence rates:__ This is a great point and we will gladly include results to that effect in the revision. Specifically, we will add estimation error convergence rates for high-dimension MI estimation via the kNN method from [15], the EDGE estimator from [16], as well as MINE [17]. In addition, convergence rates for SMI estimation via S-MINE (see Section 4.3) will be added to the current kNN-based estimate plots for completeness. Rates for these methods will be shown for several dimensions with the goal of empirically validating the scalability to high-dimensions versus lack thereof of SMI versus MI, respectively.
>
> 4. __Additional datasets:__ We will rerun the above experiment for MNIST data and expect to observe similar trends.
>
>
> 5. __RMSE definition:__ The RMSE here is the error between the SMI estimate and the true SMI; MI does not enter this definition. We will clarify this in the revision.
>
> 6. __More complex relations in Sec. 4.2:__ We will add these in revision, planning (unless the reviewer has other suggestions) to include results for sinusoidal and polynomial relationships.
>
> 7. __Discuss limitations and societal impact:__ Thank you for bringing up this important and valid point. In continuation to Item 1 above, we will add a discussion about SMI being a new measure of dependence between random variables that (although related) is different from classic MI. We will also highlight potential negative impact of treating SMI as an estimate of MI and provide the above example to support that.

---

> > ### Comment · Reviewer_C7hP · 2021-08-25
> > **Reply to Authors' response**
> >
> > Thank you for addressing all the points that I have raised.
> > I believe that the paper will become more powerful and impressive if such revisions are included.

---

### Official Review · Reviewer_imZu · 2021-07-16

**Rating:** 8
**Confidence:** 4

**Summary:**

The paper presents statistical and empirical justification for computing an average mutual information of one-dimensional slices of real-valued vector random variables. The approach avoids the curse of dimensionality in mutual information estimation by averaging over the one-dimensional slices. The resulting statistical quantity has many of the same features as mutual information and can still be used for independence testing. Additionally, the measure is not invariant to linear (or non-linear) transforms of the random variables. This means that it does not obey the information processing inequality but behaves more like a canonical correlation coefficient, but is more general:  along the 'canonical'  directions the relationship need not be linear. A dual formulation of mutual information is used in the sliced case as on objective for maximizing neural networks.  Preliminary experimental results show the potential for non-linear metric learning.

**Limitations And Societal Impact:**

I would have liked to have considered a contradictory example  (or proof that it does not happen under some regularity conditions) where where sliced mutual information approaches zero while mutual information is bounded away from zero.

**Main Review:**

**Originality** The paper adds to the literature on high-dimensional mutual information estimation with the introduction of sliced approaches that have become popular in statistical divergence measures. The approach fills a gap between non-parametric  dependence-based  estimators and canonical correlation.

**Quality** The paper's theoretical results are sound and the preliminary results illustrate the potential and utility.

**Clarity** The paper is clearly written and engaging.
Line 237 and 238 could be better reworded. "The setup in Claim 2 precludes reduction to Case 1. Indeed, if eigenvalues can shrink [...]" the subject and objects could be better labelled and incorporated into the notation used in the claim.

**Significance** High-dimensional dependence estimation and metric learning are useful for non-parametric independence testing and machine learning. The paper proposes an alternative quantity that has many useful properties while not capturing the full dependence between random variables enables estimation without the curse of dimensionality.  The approach is likely to be useful in practice.

### After author response.
I thank the authors for their response. I think it clarifies the contribution as a quantity to compute dependence without trying to estimate mutual information. I think the other reviews have brought up interesting points, especially regarding the performance in high-dimensional cases with much higher levels of mutual information (beyond the 10 class case with MNIST). I would encourage the authors to consider showcasing the work with applications of the method to post-hoc analysis of neural network models trained for tasks such as natural language modeling or image classification. In light of comments, regarding max versus mean slicing, I would encourage the reviewers to consider commenting on future extension with the optimization of a distribution of slices for each random variable, https://openreview.net/forum?id=QYjO70ACDK

**Time Spent Reviewing:**

3.5

---

> ### Author Response · Authors · 2021-08-09
> **Response to Reviewer imZu**
>
> We thank the reviewer for the constructive and positive feedback. Below we address the specific comments/concerns raised.
>
> 1. __Rewording of lines 237-238:__ We agree that the current wording is confusing and propose to replace it with the following: "The eigenvalues lower and upper bounds assumed in Claim 2 are purposed to preclude reduction to the setup of Claim 1. Indeed, if eigenvalues can shrink or grow without bound, it is always better to consider a maximizing slice (as in the RHS of (11)) than to average over several slices."
>
> 2. __Discuss limitations and societal impact:__ The reviewer has asked to address potential discrepancy between MI and SMI. This is a great point, which we will discuss in the revision. The particular case of zero SMI with strictly positive MI is impossible since $\mathsf{SI}(X;Y)=0\iff (X,Y)\mbox{ are independent}\iff \mathsf{I}(X;Y)=0$. Outside of this regime, however, the gap between MI and SMI may not be bounded. Consider the example from the beginning of Section 3.2 (note that there was a typo in the submitted version; the correct setup has $Y=X_1$ and $Z=X_2$), where for any $0<a<\infty$ we have $\mathsf{I}\big(g_a(X);Y\big)=\infty$ while $\mathsf{SI}\big(g_a(X);Y\big)$ is finite. It is thus important not to view SMI as a proxy of MI (we do not suggest it is) but rather as a measure of dependence in its own right. The goal of the theoretical exploration of SMI properties (Section 3) is to justify it as a meaningful such measure. We will add a discussion to that effect in the revision and highlight the peril of treating SMI as a proxy of MI.

---

> ### Public Comment · ~Hanqing_Yang1 · 2022-07-30
> **good paper**
>
> good paper

---

### Decision · Program_Chairs · 2021-09-28

**Decision:**

Accept (Spotlight)

**Comment:**

The paper proposes a new scalable measure for statistical independence that does not attempt to approximate standard mutual information. The proposed measure has the potential to become a very useful tool for characterising statistical (in)dependence and may thus have have broad impact.

All reviewers unanimously recommend acceptance. I agree; it is nice work.  Please incorporate the reviewers' feedback for the camera-ready version.


**Consistency Experiment:**

NeurIPS has a long history of experimentation. In 2014, NeurIPS ran an experiment in which 10% of submissions were reviewed by two independent committees to quantify the randomness in the review process. This year, we repeated a variant of this experiment to see how the quality of the review process has changed over time.  This paper was part of the experiment and was therefore assigned to two committees (consisting of reviewers, an Area Chair, and a Senior Area Chair) that reached independent decisions.  If both committees made the same recommendation, this recommendation was followed. If a single committee recommended acceptance, the paper was accepted (with the exception of a few cases in which the other committee identified what we considered a fatal flaw, e.g., an error in a key result).

This copy’s committee reached the following decision: **Accept (Spotlight)**

The other committee assigned to the paper recommended **Accept (Poster)**.  You can find the other set of reviews, along with any follow up discussion with the authors here:
https://openreview.net/forum?id=SvrYl-FDq2